# DISTRIBUTIONAL DISTANCE CLASSIFIERS FOR GOAL-CONDITIONED REINFORCEMENT LEARNING

## ABSTRACT

What does it mean to find the shortest path in stochastic environments if every strategy has a non-zero probability of failing? At the core of this question is a conflict between two seemingly-natural notions of planning: maximizing the probability of reaching a goal state and minimizing the expected number of steps to reach that goal state. Reinforcement learning (RL) methods based on minimizing the steps to a goal make an implicit assumption: that the goal is always reached within some finite horizon. This assumption is violated in practical settings and can lead to suboptimal strategies. In this paper, we bridge between these two notions of planning by estimating the probability of reaching the goal at different future timesteps. This is not the same as estimating the distance to the goal – rather, probabilities convey uncertainty in ever reaching the goal at all. We then propose a practical RL algorithm, Distributional NCE, for estimating these probabilities. Taken together, our results provide a way of thinking about probabilities and distances in stochastic settings, along with a practical algorithm for goal-conditioned RL.

## 1 INTRODUCTION

The reinforcement learning (RL) community has seen growing excitement in goal-conditioned methods in recent years. These methods promise a way of making RL self-supervised: RL agents can learn meaningful (goal-reaching) behaviors from data or interactions without reward labels. This excitement is reinforced by the fact that goal-conditioned RL also seems to suggest effective ways of learning representations that are directly aligned with the RL objective (Eysenbach et al., 2022; Ma et al., 2022). However, for a long time, there has been a sticking point in both discussion and algorithmic development of goal-conditioned RL: what is the objective?

Perhaps the most natural objective is to minimize the hitting time, the expected number of steps required to reach a goal. Indeed, this is the basis for much of the classical work in this area (often under the guise of stochastic shortest-path problems (Bertsekas & Tsitsiklis, 1991)), as well as more recent work based on dynamical distance learning (Hartikainen et al., 2019; Venkattaramanujam et al., 2019; Alakuijala et al., 2022). However, these methods implicitly assume that the goal state is always reached; without this assumption, the expected hitting time can be infinite. Nonetheless, RL researchers have proposed a number of methods to optimize this "natural" notion of distance, often with methods that first estimate this distance and then select actions that minimize this distance, methods that often achieve excellent results.

In this paper, we attempt to reconcile this tension with the steps-to-goal objective. We first lay out a few subtle issues with this objective. We show that it can lead to suboptimal behavior, both on analytic examples and on continuous-control benchmarks. What, then, is the right way to think about hitting times for goal-conditioned tasks? We advocate for taking a probabilistic approach: estimate the probability of reaching the goal after exactly $t$ steps. We extend prior work that estimates the discounted stationary distribution of future goals via contrastive learning. We do this by learning a classifier that explicitly predicts the probability of reaching the goal at specific timesteps. By estimating the probability at different values of $t$, we are able to capture the local temporal structure and thereby reason about when the goal will be reached. But, importantly, these probabilities do not assume that the goal will always be reached, i.e., these probabilities remain well-defined in settings with stochastic policies and dynamics. Our analysis shows that, in deterministic environments, these two objectives are closely related.

Based on this analysis, we propose an algorithm for goal-conditioned RL that estimates the probability of reaching the goal for varying values of $t$. Our method can be viewed as a distributional extension to recent work on contrastive RL (Eysenbach et al., 2022). Our experiments show that this framing of "distances as probabilities" offers competitive performance on both low-dimensional and image-based goal-reaching tasks. Finally, using our analysis, we propose an auxiliary objective based on a self-consistency identity that these probabilities should satisfy. Augmenting our goal-conditioned methods with this auxiliary objective can further boost performance. Taken together, our analysis not only provides a better algorithm for goal-conditioned RL, but also provides a mental model to reason about "distances" in settings with uncertainty.

## 2 RELATED WORK

**Goal-Conditioned RL.** Goal-conditioned RL is one of the long-standing problems in AI (Newell et al., 1959), and has seen much progress in recent years (Durugkar et al., 2021; Zhai et al., 2022; Andrychowicz et al., 2020; Berner et al., 2019). While many works rely on a manually-designed reward function (Andrychowicz et al., 2020; Popov et al., 2017; Akella et al., 2021; Berner et al., 2019), more recent work lifts this assumption and instead learns directly from a sparse reward (Eysenbach et al., 2022; Ghosh et al., 2019; Yang et al., 2022; Hejna et al., 2023), often using variants of hindsight relabeling (Kaelbling, 1993; Andrychowicz et al., 2017).

In the recent decade, researchers have proposed a wide array of successful approaches for goal-conditioned RL, including those based on conditional imitation learning (Sun et al., 2019; Ghosh et al., 2019; Lynch et al., 2020), temporal difference learning (Durugkar et al., 2021), contrastive learning (Eysenbach et al., 2020; 2022) and planning (Tian et al., 2021; Ma et al., 2022). Many of these approaches employ a form of hindsight relabeling (Andrychowicz et al., 2017) to improve sample efficiency, or even as a basis for the entire algorithm (Eysenbach et al., 2022). Our work builds directly on prior contrastive RL methods, which are heavily inspired by noise contrastive estimation (NCE) (Gutmann & Hyvärinen, 2010). Our key contribution will be to show how such methods can be extended to give finer-grain predictions: predicting the probability of arriving at a goal state at specific times.

**Distances in RL.** Shortest path planning algorithms are the workhorse behind many successful robotic applications, such as transportation and logistics (Kim et al., 2005; Fu & Rilett, 1998; Pattanamekar et al., 2003; Cheung, 1998). Many RL methods have built upon these ideas, such as devising methods for estimating the distances between two states (Eysenbach et al., 2019; Hartikainen et al., 2019; Venkattaramanujam et al., 2019; Alakuijala et al., 2022). Our analysis highlights some subtle but important details in how these distances are learned and what they represent, showing that distances can be ill-defined and that using distances for selecting actions can yield poor performance.

**Probabilistic approaches.** One way to look at our method is that we are learning a distributional critic to represent the likelihood of reaching the goal at each future timestep, as opposed to learning a single scalar unnormalized density over future goals (Eysenbach et al., 2020; Rudner et al., 2021). Adding this temporal dimension to the contrastive NCE (Eysenbach et al., 2022) algorithm enables the critic network to break down a complex future density distribution into hopefully simpler per-timestep probabilities. In other words, for each positive example of a state and goal, contrastive NCE receives just one bit of information (Was this goal reached?) while distributional NCE receives $\log H$ bits (When in the next H steps was this goal reached?). This framework also allows one to *(i)* enforce structural consistency for probabilities across timesteps (closely related to n-step Bellman backup), *(ii)* make the critic more interpretable, and *(iii)* reason over future probabilities as distances.

**Distributional Approaches.** Our proposed method will be reminiscent of distributional approaches to RL (Dabney et al., 2018; Bellemare et al., 2017; Sobel, 1982): rather than estimating a single scalar value, they estimate a full distribution over possible future returns. In the goal-reaching setting, it is natural to think about this distribution over future values as a distribution over distances (Eysenbach et al., 2019). However, as we will show, distances are not well defined in many stochastic settings, yet a probabilistic analogue does make theoretical sense and achieves superior empirical performance. While our proposed method does not employ temporal difference updates, Sec. 5.2 will introduce an auxiliary objective that resembles TD updates. This auxiliary objective boost performance, perhaps in a similar way that the distributional RL loss enjoys stable gradients and smoothness characteristics (Sun et al., 2022).

**Stochastic Shortest Path.** Our work is similar to prior work in stochastic shortest path problems in that we aim to learn policies that learn efficient strategies for reaching goals. However, unlike prior work in this area (Chen et al., 2021; Rosenberg et al., 2020; Tarbouriech et al., 2020), we do not assume that there always exists a policy that always succeeds in reaching the goal (i.e., a proper policy). Our objective (maximizing probabilities) will extend to the setting where even the best policy occasionally fails to reach the goal.

## 3 PRELIMINARIES

We consider the reward-free goal-conditioned RL framework, which is defined by a state-space $\mathcal{S}$, action-space $\mathcal{A}$, a transition dynamics function $p(s_{t+1} \mid s_t, a_t)$, an initial state distribution $\rho_0$ and a goal distribution $p(g)$. Unlike the classical RL framework, the reward function is implicitly defined by the transition dynamics and a discount factor $\gamma \in [0, 1) : r_g(s_t, a_t) = (1-\gamma)p(s_{t+1} = g \mid s_t, a_t)$. For this reward function, the corresponding action-value function of a goal-conditioned policy $\pi_g(a|s) = \pi(a \mid s, g)$ takes the form of the discounted future density $p^{\pi_g}(s_+ = g \mid s, a)$ over the goal states:

$$Q^{\pi_g}(s_t, a_t) = (1-\gamma)\mathbb{E}_\pi \left[ \sum_{\Delta=0}^{\infty} \gamma^\Delta p(s_{t+\Delta+1} = g \mid s_{t+\Delta}, a_{t+\Delta}) \right] = p^{\pi_g}(s_+ = g \mid s_t, a_t).$$

By using this Q-function to score actions, the policy directly maximizes the chance of reaching the goal in the future. We estimate the Q-function using noise contrastive estimation (NCE) (Gutmann & Hyvärinen, 2010), which trains a binary classifier with cross-entropy objective:

$$\arg \min_C \mathbb{E}_{g \sim p^\pi(g|s,a)}[\log C(s, a, g)] + \mathbb{E}_{g \sim p(g)}[\log(1 - C(s, a, g))],$$

where the classifier $C(s, a, g)$ learns to distinguish between samples from the distribution of future states $p^\pi(. \mid s, a)$ and the marginal goal distribution $p(g)$. The resulting Bayes' optimal classifier $C^\pi$ for a policy $\pi$ is then proportional to its Q function:

$$C^\pi(s, a, g) = \frac{p^{\pi_g}(s_+ = g \mid s, a)}{p^{\pi_g}(s_+ = g \mid s, a) + p(g)}, \quad \text{so} \quad \frac{C^\pi(s, a, g)}{1 - C^\pi(s, a, g)} = \frac{p^{\pi_g}(s_+ = g \mid s, a)}{p(g)}.$$

Since the noise distribution $p(g)$ is independent of the actions, we can then optimize a policy with respect to the classifier by $\arg \max_a C^\pi(s, a, g)$.

## 4 THE PERILS OF MONTE CARLO DISTANCE FUNCTIONS

A common strategy in prior work is to predict the number of steps that elapse between one observation and another (Tian et al., 2021; Shah et al., 2021). This estimate is then used as a distance function, either for greedy action selection (Shah et al., 2021), planning (Tian et al., 2021), or reward shaping (Hartikainen et al., 2019). We will call this approach "Monte Carlo (MC) distance regression." We define the MC distance function $d^\pi(s, a, g)$ associated with a policy $\pi$ as follows:

$$d^\pi(s, a, g) = \mathbb{E}_{\tau \sim \pi|s_i=s, a_i=a, s_{i+j}=g, j \geq i} [j - i], \tag{1}$$

where $\tau$ is a sample trajectory generated by $\pi$ that first passes through the state $s$, taking action $a$, and then $g$.

Intuitively, it seems like such an approach is performing RL with the reward function that is $-1$ at every step until the goal is reached. Prior work thus interprets the distances as a Q function. However, it turns out that this distance function is not a Q function. In this section, we show that these distance functions do not (in general) correspond to a Q function, and their predictions can be misleading.

### 4.1 TOY EXAMPLE ILLUSTRATING PATHOLOGICAL BEHAVIOR

We begin by showcasing an example to illustrate why MC regression can yield very suboptimal policies, an example which proves the following proposition.

**Proposition 1.** *Relative to the reward-maximizing policy, MC regression can incur regret that is arbitrarily large. That is, for any regret $R \in \mathbb{R}$, there exists an MDP with goal state $g$ and*

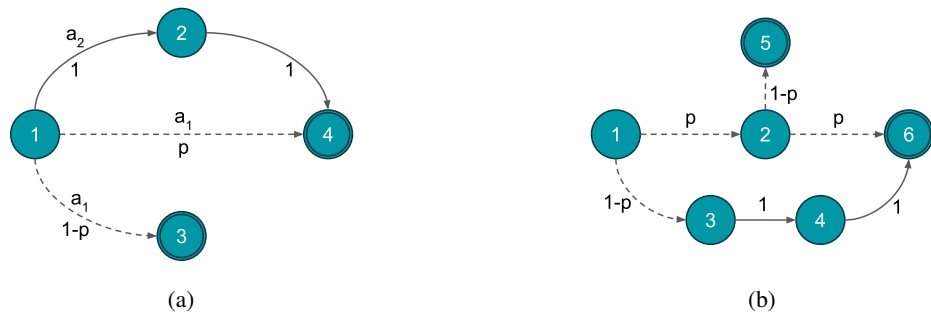

(a)                 (b)

Figure 1: Toy MDPs to illustrate the pathological behaviors exhibited by MC distance regression. Solid lines and dashed lines denote deterministic and stochastic state transitions, respectively.

*reward function $r_g(s, a) = \mathbb{1}(s = g) - 1$ such that the difference in expected returns between the return-maximizing policy and a policy learned via MC regression ($\pi_{MC}$) is at least R:*

$$\max_\pi \mathbb{E}_\pi \left[ \sum_{t=0}^\infty \gamma^t r_g(s_t, a_t) \right] - \mathbb{E}_{\pi_{MC}} \left[ \sum_{t=0}^\infty \gamma^t r_g(s_t, a_t) \right] \geq R.$$

*Proof.* We prove this proposition by constructing such an MDP (see Fig. 1(a)). The goal state is **4** and the absorbing state is **3**. From state **1**, the agent can choose an action $a_1$ to directly reach the goal state **4** in a single step with a probability of $p$, but risks getting trapped in state **3** with $1 - p$ odds. On the other hand, the agent can choose an action $a_2$ to deterministically reach the goal **4** in 2 steps. The agent receives a reward of $-1$ at every timestep it is not at the goal, and the episode terminates once the agent reaches the goal state **4**.

For this example, let's estimate the MC distance function $d(s, a, g)$ using the definition in Eq. 1. Interestingly, $d(1, a_1, 4) = 1$. This is because all the rollouts that start from the state **1** and *reach the goal state* **4** after taking an action of $a_1$ are always unit length. Similarly, $d(1, a_2, 4) = 2$. If we treat $-d(s, a, g)$ as the Q function, we will always end up picking action $a_1$.

Assuming a discount factor $\gamma$, we can compute the optimal Q function analytically: $Q(1, a_1, 4) = -\frac{(1-\gamma p)}{(1-\gamma)}$ and $Q(1, a_2, 4) = -(1+\gamma)$. When the transition probability $p < \gamma$, choosing the action $a_1$ is suboptimal with a linear regret of $Q(1, a^* = a_2, 4) - Q(1, a_1, 4) = \frac{\gamma(\gamma-p)}{(1-\gamma)}$. In the limit $\gamma \to 1$, this regret is unboundedly large for any $p \in [0, 1)$ (proves Proposition 1). $\qquad\square$

The above example highlights that MC distances ignore the risk of getting indefinitely stuck in the trap state **3**. Moreover, the MC distance does not depend on the transition probability $p$, suggesting that it offer an optimistic distance estimate by ignoring the stochasticity in dynamics. Acting greedily with an MC distance function results in a policy that takes the shortest path on the graph by treating stochastic edges as being deterministic, which can be very suboptimal in stochastic settings. For instance, Fig. 2 shows that if the transition probability $p = 0.1$ for $\mathbf{1} \to \mathbf{4}$, MC distance suggests the suboptimal action $a_1$ which incurs a significantly higher regret that the optimal action $a_2$, as suggested by the optimal Q-function. This demonstrates a fundamental disconnect between shortest-path solutions and reasoning about the likelihood of reaching a goal state in the future.

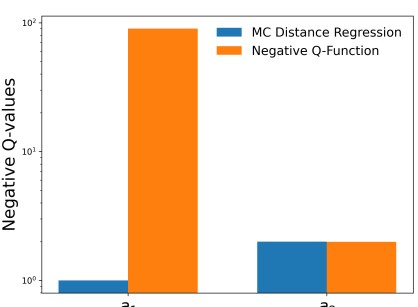

Figure 2: MC distances and optimal negative Q-values at disagreement for $\mathbf{1} \to \mathbf{4}$ on the toy MDP in Fig. 1(a) with $\gamma = 0.99$ and $p = 0.1$. The y-axis has a logarithmic scale.

**Proposition 2.** *There exists an MDP such that the the "distance" learned by MC regression $d : \mathcal{S} \times \mathcal{S} \to \mathbb{R}^+$ violates the triangle inequality:*

$$d(s_1, s_3) \geq d(s_1, s_2) + d(s_2, s_3), \; \exists\, s_1, s_2, s_3 \in \mathcal{S}.$$

*Proof.* Proof by contradiction: we prove Proposition 2 by constructing a toy MDP where the MC distance function violates the triangle inequality. Consider the MDP in Fig. 1b, where the agent has no control over state transitions through actions. The MC distance function $d(s, g)$[1] answers the following question: *if the agent traveled from $s$ to $g$, how many steps would elapse (on average)?*. For example, $d(\mathbf{3}, \mathbf{4}) = 1$ because this state $\mathbf{4}$ always occurs one step after state $\mathbf{3}$. However, $d(\mathbf{1}, \mathbf{2}) = 1$: even though it may be unlikely that state $\mathbf{2}$ occurs after state $\mathbf{1}$, *if* state $\mathbf{2}$ occurs, it would occur after a single step. Similarly, $d(\mathbf{2}, \mathbf{6}) = 1$. While $d(\mathbf{1}, \mathbf{2}) + d(\mathbf{2}, \mathbf{6}) = 2$, the estimated MC distance directly from $\mathbf{1}$ to $\mathbf{6}$ is $d(\mathbf{1}, \mathbf{6}) = \frac{2p^2 + 3(1-p)}{p^2 + (1-p)}$ is greater than 2 for all $p \in [0, 1)$. This is a violation of triangle inequality, since $d(\mathbf{1}, \mathbf{6}) > d(\mathbf{1}, \mathbf{2}) + d(\mathbf{2}, \mathbf{6})$. □

**Corollary 1.** There exists an MDP such that the "distance" learned by MC regression $d : \mathcal{S} \times \mathcal{S} \rightarrow \mathbb{R}^+$ is (i) not a proper metric or quasimetric and (ii) not an optimal goal-conditioned value function for any non-negative cost function.

*Proof.* Corollary 1 directly follows from the proof of Proposition 1. MC distance functions do not always satisfy the triangle inequality and hence cannot be a valid distance metric (nor quasimetric). Moreover, an optimal goal-conditioned value function has to obey the triangle inequality (Wang et al., 2023). Thus, the MC distance function cannot correspond to the optimal goal-conditioned value function for any non-negative cost function. □

**Why do MC distance functions exhibit these pathological behaviors?** In the examples from Fig. 1, the MC distance estimates do not account for the transitions that could result in getting stuck in a trap state ($\mathbf{3}$ and $\mathbf{5}$ in Fig. 1(a) and (b) respectively). More generally, the pathological behaviors of MC distances can be attributed to their optimism bias, wherein they are computed assuming the agent will inevitably reach the goal without considering the associated risks.

## 4.2 CONNECTION BETWEEN MAXIMIZING LIKELIHOOD AND STOCHASTIC SHORTEST PATH

Imagine an MDP where the episode does not terminate upon reaching the goal. In this setting, the reward-free goal-conditioned RL agent that is incentivized to maximize its time at the goal is closely related to an agent that is trying to minimize the expected time to the goal (proof in Appendix A):

$$\max_{\pi} \log \left( (1 - \gamma) \mathbb{E}_{\pi} \left[ \sum_{t=0}^{\infty} \gamma^t r(s_t, a_t, g) \right] \right) \geq \max_{\pi} -\mathbb{E}_{\Delta \sim \pi}[\Delta] \log \left( \frac{1}{\gamma} \right),$$

where $\Delta \sim \pi$ denotes the length of a trajectory drawn out of the policy $\pi$ to reach the goal.

*If both maximizing likelihood and shortest-path planning seem closely related in theory, why do shortest-path methods suffer from pathological behaviors?* The answer lies in the logarithmic transformation that gets applied to the likelihood. In simple words, the likelihood of success while failing to reach the goal is 0, which is a well-defined number, whereas the corresponding expected distance to the goal is unboundedly large (the negative logarithm of 0).

## 4.3 INTERPRETING DISTANCE REGRESSION AS A CLASSIFICATION PROBLEM

Monte-Carlo distance regression can be estimated by learning a *normalized* distance classifier over the observed horizon, followed by using the bin probabilities to obtain the mean distance. More precisely, let $H \in \{0, 1, \cdots B - 1\}$ be a random variable denoting how far ahead to look to sample the future states (a.k.a goals). The distance classifier represented by $C(s, a, g) \in \mathcal{P}^B$ can then be learned using a categorical cross-entropy loss:

$$\mathbb{E}_{p(H), s_t, a_t \sim p(s,a), g \sim p^{\pi}(s_{t+H}|s_t, a_t)} \left[ \log C(s_t, a_t, g)[H] \right].$$

Obtaining distances from this classifier is straightforward: $d(s, a, g) = \sum_H H \, C(s, a, g)[H]$. Using Bayes' Rule, we can express the Bayes optimal classifier as

$$C^{\pi}(s_t, a_t, g)[H] = P^{\pi_g}(H \mid s_t, a_t, g) = \frac{p^{\pi_g}(s_{t+H} = g \mid s_t, a_t) p(H)}{p^{\pi_g}(s_+ = g \mid s, a)}. \tag{2}$$

---

[1]$d(s, g)$ is short for $d(s, a = \varnothing, g)$ in a Markov process (i.e., an MDPs without actions). Fig. 1b is one such example.

This expression reveals a subtle nuance with distance regression. This distance classifier predicts normalized probabilities, which implicitly assume that the goal can be reached within a finite horizon. Consider this example: say that action $a_1$ has $p^{\pi_g}(g \mid s, a_1, H = 1, 2, 3, ...) = [0.01, 0, 0, \cdots]$ while $a_2$ has $p^{\pi_g}(g \mid s, a_2, H = 1, 2, 3, ...) = [1, 1, 1, \cdots]$. Then, the normalized distance classifier (Eq. 2) prefers action $a_1$ over $a_2$ since $d(s, a_1, g) = 1$ and $d(s, a_2, g) > 1$, despite it succeeding in reaching the goal with $100\times$ lower probability. On the other hand, when goal $g$ is unreachable from $(s, a)$, i.e., $C(s, a, g)[H] = [0, 0, ...]$, the MC distance is ill-defined, as it results in division by 0.

## 5 THE FIX: ESTIMATE PROBABILITIES, NOT DISTANCES

In this section, we propose a method that directly estimates the probabilities of reaching goals at different horizons. We describe our method and provide analysis in Sec. 5.1. As we will show in our experiments, this method can already achieve excellent results in its own right. Sec. 5.2 proposes a regularization term based on an identity that our probabilities should satisfy. Our experiments will demonstrate that adding this regularization term can further boost performance.

### 5.1 OUR METHOD: DISTRIBUTIONAL NCE

The underlying issue with distance classifiers (discussed in Sec. 4.3) is that they are normalized across the horizon; they have a softmax activation. Replacing that softmax activation with a sigmoid activation resolves this issue and opens the door to new algorithms that resemble distributional RL.

The connection with distributional RL is interesting because it motivates distributional RL in a different way than before. Usually, distributional RL is motivated as capturing aleatoric uncertainty, providing information that can disambiguate between a strategy that always gets +50 returns and a strategy that gets +100 returns 50% of the time. Here, we instead show that distributional RL emerges as a computationally efficient way of learning distances, not because it gives us any particular notion of uncertainty. This is also interesting in light of prior work that distributional RL does not necessarily produce more accurate value estimates (Bellemare et al., 2017).

We start by introducing an MC method to learn a distance classifier $C(s, a, g) \in [0, 1]^B$; note that each element of this vector is a probability, but they need not sum up 1. This distance classifier can be learned via *binary* classification:

$$\max_C \mathbb{E}_{p(H)p(s_t, a_t)} \left[ \mathbb{E}_{g \sim p^\pi(s_{t+H}|s_t, a_t)}[\log C(s_t, a_t, g)[H]] + \mathbb{E}_{p(g)}[\log(1 - C(s_t, a_t, g)[H])] \right].$$
(3)

The Bayes' optimal classifier satisfies

$$\frac{C^\pi(s_t, a_t, g)[H]}{1 - C^\pi(s_t, a_t, g)[H]} = \frac{p^{\pi_g}(s_{t+H} = g \mid s_t, a_t)}{p(g)}.$$
(4)

On the RHS, note that actions only appear in the numerator. This means that selecting the actions using the LHS is equivalent to selecting the actions that maximize the probability of getting to the goal in exactly $H$ steps. While this notion of success is non-Markovian, this same classifier can be used to maximize the (Markovian) RL objective with $r(s, a, g) = \mathbb{1}(s = g)$ using the following:

$$\sum_{\Delta=1}^\infty \gamma^{\Delta-1} \frac{C^\pi(s_t, a_t, g)[\Delta]}{1 - C^\pi(s_t, a_t, g)[\Delta]} = \sum_{\Delta=1}^\infty \gamma^{\Delta-1} \frac{p^{\pi_g}(s_{t+\Delta} = g \mid s_t, a_t)}{p(g)} = \frac{p^{\pi_g}(s_+ = g \mid s_t, a_t)}{(1 - \gamma)p(g)}. \quad (5)$$

The expression on the RHS is the same as the objective in Contrastive NCE (Eysenbach et al., 2022), which corresponds to maximizing the likelihood of the goal state under the discounted state occupancy measure. However, a regular contrastive critic only receives one bit of information (Was this goal reached?) from a positive example of a state and goal, whereas a distributional critic receives $\log H$ bits (When in the next H steps was this goal reached?). This additional supervision may help explain why, empirically, distributional NCE outperforms Contrastive NCE in practice (Fig. 4, Sec. 6).

In practice, we use the last bin of the distributional NCE classifier as a catch-all bin. This modification avoids ill-defined Q-values due to a finite number of bins, by accounting for the future states from the trajectory that are at least $h$ steps away, where $h$ is the number of classifier bins in the distributional NCE algorithm. See the Appendix B.1 for more details about using the catch-all bin. Implementing the distributional NCE fix is easy: (1) change the final activation of the distance classifier from a softmax to a sigmoid; (2) change the loss for the distance classifier from a categorical cross-entropy to an (elementwise) binary cross entropy.

---

**Algorithm 1** DISTRIBUTIONAL NCE: `h` is the number of bins in the classifier output, which may be less than the task horizon. Comments denote the shapes of tensors.

---

```python
def critic_loss(states, actions, future_states, dt):
    # dt: relative time index of future state
    logits = classifier(states, actions, future_states)  # (batch_size, batch_size, h)
    probs = sigmoid(logits)
    labels = one_hot(dt, num_classes=h)
    loss = BinaryCrossEntropy(logits, labels)
    return loss.mean()

def actor_loss(states, goals):
    actions = policy.sample(states, goal=goals)   # (batch_size, action_dim)
    logits = classifier(states, actions, goals)   # (batch_size, batch_size, h)
    prob_ratio = exp(logits) # p(g|s,a,h) / p(g) = C(s,a,g)[h] / (1 - C(s,a,g)[h])
    Q = sum(discount ** range(h) * prob_ratio, axis=-1)   # (batch_size, batch_size)
    return -1.0 * Q.mean()
```

---

**Analysis.** The Bayes optimal MC distance classifier can be obtained from normalizing the Bayes optimal distributional NCE classifier across the horizon:

$$P^{\pi_g}(H = h \mid s_t, a_t, g) = \frac{p^{\pi_g}(s_{t+h} = g \mid s_t, a_t)P(h)}{p^{\pi_g}(s_+ = g \mid s_t, a_t)} = \frac{w^{\pi}(s_t, a_t, g)[h]P(h)}{\sum_{h'} w^{\pi}(s_t, a_t, g)[h']P(h')}, \quad (6)$$

where $w^{\pi}(s, a, g)[h] = \frac{C^{\pi}(s,a,g)[h]}{1 - C^{\pi}(s,a,g)[h]}$. The Q-function we obtain from aggregating the bins of the distributional NCE classifier with geometric weights (Eq. 5) is the same as the contrastive NCE method (Eysenbach et al., 2022). Under mild assumptions (invoking the results from Sec. 4.5 and Appendix B in Eysenbach et al. (2022)), we prove that distributional NCE is performing approximate policy improvement and is a convergent contrastive RL algorithm (more details in Appendix B.2).

## 5.2 SELF-SUPERVISED TEMPORAL CONSISTENCY OBJECTIVE

The problem of learning goal-directed behavior exhibits a certain structure: the probability of reaching a goal in 10 days starting today is related to the probability of reaching that same goal in 9 days starting tomorrow. In Appendix C, we derive the following auxiliary objective based on this idea:

$$L_{TC}^k = \mathbb{E}_{(s_t, a_t, g, s_{t+k}, a_{t+k})}\Big[ \lfloor C(s_{t+k}, a_{t+k}, g)[H - k] \rfloor \log C(s_t, a_t, g)[H]$$
$$+ \lfloor (1 - C(s_{t+k}, a_{t+k}, g)[H - k]) \rfloor \log \big(1 - C(s_t, a_t, g)[H]\big) \Big]. \quad (7)$$

We hypothesize that adding this additional objective to distributional NCE will enable information to flow back in time and accelerate training.

## 6 EXPERIMENTS

In this section, we provide empirical evidence to answer the following questions:

1. Does the distributional NCE algorithm offer any benefits over the MC distance regression and distance classifier in deterministic goal-reaching environments, with function approximation and a stochastic policy?
2. Can distributional NCE accurately estimate the probability of reaching the goal at a specific future time step?
3. Are there any benefits to using the distributional architecture for classifier learning?
4. Does the temporal consistency term accelerate the distributional NCE training?

**Environments.** We selected seven standard goal-conditioned environments (Plappert et al., 2018; Yu et al., 2020) to test these hypotheses: fetch_reach, fetch_push, sawyer_push, sawyer_bin, fech_reach_image, fetch_push_image, and sawyer_push_image. The latter three environments have image-based observations. fetch_reach is the simplest task; The fetch_push and sawyer_push environments are more challenging and require the robot to use its gripper to push an object to the specified goal position. Lastly, the pick-and-place in sawyer_bin presents a hard exploration challenge. See Appendix E for more information on the environments and implementation details.

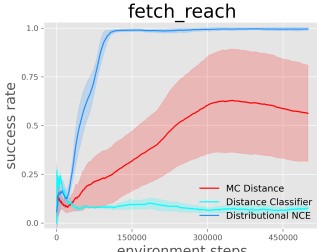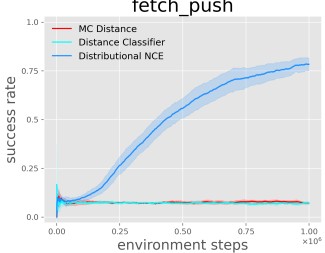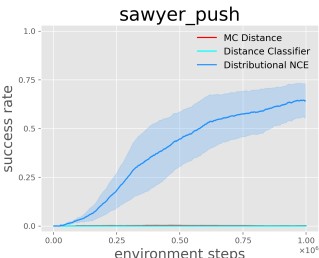

Figure 3: Distributional NCE is able to solve all the goal-reaching tasks with a good success rate, whereas the MC distance functions fail at almost all the tasks, despite the tasks being deterministic. This performance degradation is also seen in stochastic environments as shown in Appendix D.4. This result supports our hypothesis that MC distances are not a good choice for the Q-function of goal-conditioned RL tasks.

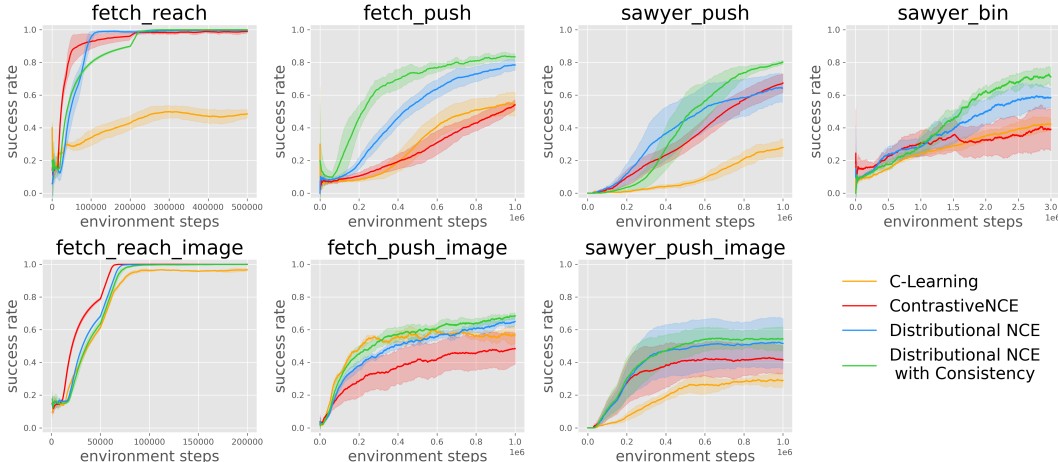

Figure 4: **Comparison with baselines.** Distributional NCE outperforms the Contrastive NCE (Eysenbach et al., 2022) and C-Learning (Eysenbach et al., 2020) in all but the easiest tasks (fetch_reach, fetch_reach_image). Applying temporal consistency on top of Distributional NCE accelerates learning and boosts asymptotic performance. More comparisons with prior GCRL baselines can be found in Appendix D.1.

**Comparison with distance regression.** In the earlier section, we showed that using the MC distance metric can be very suboptimal for stochastic MDPs with a countable state space, where the optimal policy was known beforehand. Our first experiment is designed to test if distance functions learned via MC regression and distance classifier can be used in the place of a Q-function to greedily optimize a stochastic policy. We hypothesize that the stochasticity in action sampling from the policy, along with the associated risk of choosing the shortest path are ignored by MC distance functions, which will result in suboptimal behavior. We test out the MC distance regression and distance classifier algorithms on the three following tasks with increasing difficulty: fetch_reach, fetch_push, and sawyer_push. We also included a comparison with distributional NCE to check if the proposed algorithm fills in the shortcomings of using MC distance functions. We use the same number of classifier bins for both the distance classifier and the distributional NCE.

Our results from Fig. 3 suggest that MC distance regression only succeeds at fetch_reach, the simplest of the selected tasks, which only requires greedily moving to a target goal position. Surprisingly, MC distance classifier fails at all the tasks. In every other setting, MC distance functions are not able to do considerably better than a randomly initialized policy. On the other hand, the distributional NCE algorithm is able to learn a policy that solves all the tasks.

**Comparing to prior goal-conditioned RL algorithms.** We now compare the performance of distributional NCE against two high-performance goal-conditioned RL algorithms: Contrastive NCE and C-learning algorithms. Comparing against Contrastive NCE directly allows us to study whether our distributional critic boosts performance, relative to a contrastive method (contrastive NCE) that predicts a single scalar value. Distributional NCE is an on-policy algorithm, so the comparison with C-learning (an off-policy algorithm) lets us study whether this design decision decreases performance.

The results, shown in Fig. 4, demonstrate that distributional NCE is roughly on par with the prior methods on the easiest tasks (fetch_reach and fetch_reach_image), but can perform notably better on

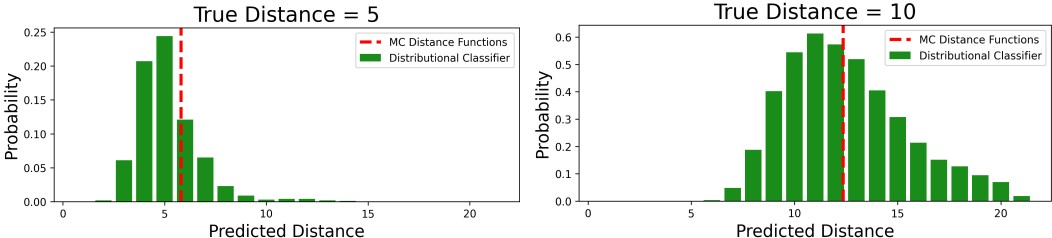

Figure 5: Visualizing the probabilistic distance predictions for future goals that are 5 *(Left)* and 10 *(Right)* steps away, on the fetch_push task. These results confirm that the distance predictions offered by distributional NCE correlate well with the true distance and are well-calibrated in uncertainty.

some of the more challenging tasks; relative to the strongest baseline, distributional NCE realizes a $+24\%$ improvement on fetch_push and a $+20\%$ improvement on sawyer_bin.

As discussed in Sec. 5.2, the predictions from distributional NCE should obey a certain consistency property: the probability of getting to a goal after $t$ time steps from the current state should be similar to the probability of getting to that same goal after $t-1$ steps starting at the next state. We equip distributional NCE with the auxiliary objective proposed in Eq. 7 based on this property. We show the results from this variant of distributional NCE ("distributional NCE with consistency") in green in Fig. 4. While we see no effect on the easiest tasks (fetch_reach, fetch_reach_image), the auxiliary term improves the sample efficiency of the fetch_push task ($2.8 \times 10^5$ fewer samples to get to 60% success rate) and improves the asymptotic performance on the sawyer_push and sawyer_bin tasks by $+16\%$ and $+13\%$ respectively.

**Analyzing distributional NCE's predictions.** To better understand the success of distributional NCE, we visualize its predictions. We do this by taking two observations from the fetch_reach task that take 5 steps to transit between under a well-trained policy. We show the predictions from distributional NCE in Fig. 5 *(left)*. Note that distributional NCE outputs a probability for each time step $t$. The highest probability is for reaching the goal after exactly 5 steps, but the method still predicts that there is a non-zero probability of reaching the goal after 4 steps or after 6 steps. We also compare to the predictions of the "MC Distance" baseline from Fig. 3. We see that this baseline makes an accurate estimate for when the goal will be reached.

We include another visualization of the distributional NCE predictions in Fig. 5 *(right)*, this time for two observations that occur 10 steps apart. Again, the predictions from distributional NCE appear accurate: the goal has the highest probability of being reached after $10 - 12$ steps. These predictions highlight an important property of the distributional NCE predictions: they do not sum to one. Rather, it may be likely that the agent reaches the goal after 9 steps and remain at that goal, so the probability of being in that goal after 11 steps is also high.

## 7 CONCLUSION

This paper takes aim at the tension between two conflicting objectives for goal-reaching: maximizing the probability of reaching a goal, and minimizing the distance (number of steps) to reach a goal. Our analysis shows that distance-based objectives can cause poor performance on both didactic and benchmark tasks. Based on our analysis, we propose a new method that predicts the probability of arriving at the goal at many different time steps; this method outperforms prior goal-conditioned RL methods, most notably those based on regressing to distances. Our analysis also suggests a temporal-consistency regularizer, which can be added to boost performance. Together, we believe that these results may prove hopeful both to new researchers attempting to build a mental model for goal-conditioned RL, as well as veteran researchers aiming to develop ever more performant goal-conditioned RL algorithms.

**Limitations.** One limitation of our method, compared with prior contrastive approaches, is that the classifier is now tasked with predicting many values (one per time step) rather than a single value. We use the same architecture as the contrastive NCE (Eysenbach et al., 2022) baseline while changing the output dimension of the last linear layer in the critic network. While this increases the number of parameters ($+6.5\%$), we found it had a negligible effect on training speed. A second limitation is that our method is on-policy: it estimates the probabilities of reaching goals under the behavioral policy. Figuring out how to build performant goal-conditioned algorithms that can perform off-policy, distributional reasoning remains an important problem for future work.

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

# A CONNECTION BETWEEN MAXIMIZING LIKELIHOOD AND STOCHASTIC SHORTEST PATH METHODS

In this section, we provide proof for the statements in Sec. 4.2. Let's consider a policy $\pi$ for reaching a goal and define $\Delta$ as the number of timesteps required to reach the goal from the start state. Note that $\Delta$ is a discrete random variable that takes integer values. Formally, we define $\pi(\Delta)$ as the distribution over the number of timesteps required to reach the goal under the policy. Then, a policy that tries to reach the goal as soon as possible is trying to optimize the following objective:

$$\max_\pi -\mathbb{E}_{\Delta \sim \pi}[\Delta] \tag{8}$$

Alternatively, consider an MDP where the episode does not terminate upon reaching the goal. In this setting, the reward-free goal-conditioned RL agent is incentivized to maximize its time at the goal:

$$\max_\pi (1-\gamma)\mathbb{E}_\pi \left[ \sum_{t=0}^\infty \gamma^t r(s_t, a_t, g) \right] = \max_\pi (1-\gamma)\mathbb{E}_\pi \left[ \sum_{t=0}^\infty \gamma^t \delta(s_t == g) \right]$$

$$= \max_\pi (1-\gamma)\mathbb{E}_{\Delta \sim \pi} \left[ \sum_{t=0}^\infty \gamma^{t+\Delta} \right]$$

$$= \max_\pi (1-\gamma)\mathbb{E}_{\Delta \sim \pi} \left[ \frac{\gamma^\Delta}{1-\gamma} \right]$$

$$= \max_\pi \mathbb{E}_{\Delta \sim \pi} \left[ \gamma^\Delta \right].$$

By applying a $\log$ transformation on both sides of the equation, followed by Jensen's inequality, we get:

$$\max_\pi \ \log \left( (1-\gamma)\mathbb{E}_\pi \left[ \sum_{t=0}^\infty \gamma^t r(s_t, a_t, g) \right] \right) = \max_\pi \ \log \left( \mathbb{E}_{\Delta \sim \pi} \left[ \gamma^\Delta \right] \right)$$

$$\geq \max_\pi \ \mathbb{E}_{\Delta \sim \pi} \left[ \log \left( \gamma^\Delta \right) \right]$$

$$= \max_\pi \ -\mathbb{E}_{\Delta \sim \pi}[\Delta] \log \left( \frac{1}{\gamma} \right). \tag{9}$$

The final RHS expression can be interpreted as minimizing the expected time to the goal under the policy (Eq. 8), which corresponds to the shortest-path planning objective with a discount factor of 1. Thus, optimizing the shortest-path planning objective with a discount factor of 1 is a lower bound to the max likelihood objective with a discount factor $\gamma < 1$. If the policy always takes the same number of steps to reach the goal, i.e. $\pi(\Delta)$ is a Dirac distribution, then the lower bound becomes an equality and maximizing the probability of reaching the goal (LHS) is equivalent to minimizing the expected steps to reach the goal (RHS). One setting where this always happens is deterministic MDPs with deterministic policies.

*If both maximizing likelihood and shortest-path planning seem closely related in theory, why do shortest-path methods suffer from pathological behaviors?* The answer lies in the logarithmic transformation that gets applied to the likelihood. In simple words, the likelihood of success while failing to reach the goal is 0, which is a well-defined number, whereas the corresponding expected distance to the goal is unboundedly large (the negative logarithm of 0). More formally, the problem with optimizing the shortest-path objective in RHS is that it remains unclear how to correctly train a distance function in stochastic settings, when every strategy has a non-zero chance of failing to reach the goal. For instance, training a distance function via MC regression (Hartikainen et al., 2019; Tian et al., 2021) provides optimistic distance estimates because the training goals are always reached within some finite horizon, which can result in very sub-optimal behaviors as shown in Sec. 4.1. A naive approach to fixing this optimism bias is to train the distance function on unreachable goals as well. However, this poses two practical problems:

1. *Sampling from the distribution of unreachable goals under a policy is non-trivial*: One can sample from the set of easily reachable goals (positive examples) under the policy by simply rolling it out in the environment for a short duration. However, sampling far-away goals

> (hard to reach under the current policy) requires one to run long policy rollouts, making such far-away goals sparser than easily reachable goals in the collected dataset. Extending this idea to the limit, one can simply never know if a state is unreachable from the policy even after a large number of steps through Monte-Carlo policy rollouts alone. But even if one could sample these negative goals,

2. *optimal distance functions become ill-defined when the regression targets are unboundedly large*: An unreachable state has an unboundedly large target distance (infinity). This makes it numerically unstable to perform direct MC regression since a part of the dataset involves regressing to infinite target distances. Alternatively, one can learn a *normalized* distance classifier (Sec. 4.3) with a catch-all bin to handle hard-to-reach and unreachable goals. However, converting such a distance classifier into an MC distance function by computing the expected distance $d(s, a, g) = \sum_H H \ C(s, a, g)[H]$ is again ill-defined since the upper-bound of the catch-all bin is unboundedly large (infinity).

Prior works in contrastive RL (Eysenbach et al., 2022; 2020) are closely related to the former idea of sampling negative goal examples with subtle modifications: (1) instead of sampling from the distribution of unreachable goal states, we simply sample from a noise distribution, and (2) replace regression objective with the NCE classification objective (Gutmann & Hyvärinen, 2010) to differentiate between samples drawn from the positive and negative goal distributions. However, these methods directly estimate the likelihood of reaching the goal without providing any information about the dynamical distance, i.e., the expected timesteps to reach the goal. Our work proposes a distributional variant of contrastive NCE algorithm (Eysenbach et al., 2022), which can: (1) estimate the likelihood of reaching the goal, and (2) reason about the dynamical distance via normalization using Bayes rule (Eq. 6).

## B  ANALYSIS OF THE DISTRIBUTIONAL NCE ALGORITHM

In this section, we introduce the modifications to the Distributional NCE framework from Sec. 5.1 to turn it into a practical algorithm. We start by introducing a catch-all bin in B.1 to avoid truncation errors and optimize for the true (Markovian) RL objective. Next, we provide convergence guarantees for the Distributional NCE Algorithm in B.2, by drawing equivalence to a prior convergent contrastive RL algorithm. Lastly, we provide the derivation for 1-step and Multi-step temporal consistency regularization in C, highlighting their connections to temporal difference (TD) learning approaches.

### B.1  NECESSITY OF A CATCH-ALL BIN

In Sec. 5.1, we introduced the Distributional NCE algorithm (Alg. 1) that estimates the likelihood of reaching the goal at specific future timesteps (up to proportionality, Eq. 4). We then showed that these estimates can be aggregated using geometrically-decaying weights to optimize for the (Markovian) RL objective with $r(s, a, g) = \mathbb{1}(s = g)$ in Eq. 5. However, implementing this naively would require a large number of bins to prevent temporal truncation errors and could lead to ill-defined Q-values.

The contrastive learning framework (Gutmann & Hyvärinen, 2010) used in Distributional NCE and prior works (Eysenbach et al., 2022; 2020) can estimate any arbitrary positive goal distribution upto a proportionality, as long as one can draw samples from it. In the Distributional NCE implementation with $h$ classifier bins, we repurpose the last bin to predict if the goal was sampled for $t \geq h$ rather than $t == h$ event, referring to it as the "catch-all" bin in the rest of the paper. More precisely, the objective for the catch-all bin is as follows:

$$\max_C \mathbb{E}_{p(H \geq h)p(s_t, a_t)} \left[ \mathbb{E}_{g \sim p^\pi(s_{t+H}|s_t, a_t)}[\log C(s_t, a_t, g)[h]] + \mathbb{E}_{p(g)}[\log(1 - C(s_t, a_t, g)[h])] \right],$$

(10)

where $p(H \geq h) = (1 - \gamma)\gamma^{H-h} = \text{GEOM}(\gamma)[H - h]$ is a Geometric distribution shifted by $h$ units. Then, the Bayes' optimal catch-all classifier for a policy $\pi$ satisfies:

$$\frac{C^\pi(s_t, a_t, g)[h]}{1 - C^\pi(s_t, a_t, g)[h]} = \mathbb{E}_{p(H \geq h)} \left[ \frac{p^{\pi_g}(s_{t+H} = g \mid s_t, a_t)}{p(g)} \right].$$

(11)

This classifier can be used to maximize the (Markovian) RL objective with $r = \mathbb{1}(s = g)$ as follows:

$$\sum_{\Delta=1}^{h-1} \left( (1-\gamma)\gamma^{\Delta-1} \frac{C^\pi(s_t, a_t, g)[\Delta]}{1 - C^\pi(s_t, a_t, g)[\Delta]} \right) + \gamma^{h-1} \frac{C^\pi(s_t, a_t, g)[h]}{1 - C^\pi(s_t, a_t, g)[h]}$$

$$= \sum_{\Delta=1}^{h-1} \left( (1-\gamma)\gamma^{\Delta-1} \frac{p^{\pi_g}(s_{t+\Delta} = g | s_t, a_t)}{p(g)} \right) + \gamma^{h-1} \mathbb{E}_{p(H \geq h)} \left[ \frac{p^{\pi_g}(s_{t+H} = g \mid s_t, a_t)}{p(g)} \right]$$

$$= \sum_{\Delta=1}^{h-1} \left( (1-\gamma)\gamma^{\Delta-1} \frac{p^{\pi_g}(s_{t+\Delta} = g | s_t, a_t)}{p(g)} \right) + \gamma^{h-1} \sum_{\Delta=h}^{\infty} \left( (1-\gamma)\gamma^{\Delta-h} \frac{p^{\pi_g}(s_{t+\Delta} = g | s_t, a_t)}{p(g)} \right)$$

$$= (1-\gamma) \sum_{\Delta=1}^{\infty} \left( \gamma^{\Delta-1} \frac{p^{\pi_g}(s_{t+\Delta} = g | s_t, a_t)}{p(g)} \right)$$

$$= \frac{p^{\pi_g}(s_+ = g | s_t, a_t)}{p(g)} \tag{12}$$

The expression on the RHS is the same as the objective in C-learning, which corresponds to maximizing the likelihood of the goal state under the discounted state occupancy measure.

In Distributional NCE, each classifier bin is crucial for estimating the corresponding component in the discounted future state density. An interesting future direction can be to employ redundancy in classifier bins, i.e., use multiple catch-all bins and exploit the relation between them as additional temporal consistency. Such a temporal ensembling procedure can be very similar to consistency training approaches (Xie et al., 2020) from semi-supervised learning literature.

### B.2 CONVERGENCE PROOF

To prove convergence of the Distributional NCE algorithm, we make the same assumptions as the Contrastive NCE (Eysenbach et al., 2022) work:

1. *Bayes-optimality of the Critic*: We assume that the distributional critic is Bayes-optimal for the current policy.

2. *Training Data Filtering*: We only consider $(s_t, a_t, s_{t+h})$ tuples for the policy improvement step, whose probability of the trajectory $\tau_{t:t+h} = (s_t, a_t, s_{t+1}, a_{t+1}, ..., s_{t+h})$ when sampled from $\pi(.|., s_g)$ under the commanded goal $s_g$ is close to the probability of the same trajectory when sampled from $\pi(.|., s_{t+h})$, under the relabelled goal $s_{t+h}$.

**Proposition 3.** *When the above-mentioned assumptions hold, the Distributional NCE update corresponds to approximate policy improvement in tabular settings.*

*Proof.* We first point out that the Bayes optimal Distribuitional NCE critic can be used to obtain the Bayes optimal Contrastive NCE (Eysenbach et al., 2022) critic, by geometrically averaging the classifier bins according to Eq. 12. Using this result, Proposition 3 is validated by the proof for the Contrastive NCE update corresponding to approximate policy improvement in tabular settings (Sec 4.5 and Appendix B in Eysenbach et al. (2022)). This result still holds when we apply the consistency objective, since the Bayes optimal distributional critics are temporally consistent (Eq. 17). □

## C TEMPORAL CONSISTENCY AUXILIARY OBJECTIVE

The problem of learning goal-directed behavior exhibits a certain structure: the probability of reaching a goal in 10 days starting today is related to the probability of reaching that same goal in 9 days starting tomorrow. This idea highlights a simple identity that the distributional probabilities must satisfy to remain temporally consistent:

$$p^{\pi_g}(g \mid s_t, a_t, H) = \mathbb{E}_{(s_{t+1}, a_{t+1}) \sim (s_t, a_t)} \left[ p^{\pi_g}(g \mid s_{t+1}, a_{t+1}, H-1) \right].$$

Note that the probabilities on both LHS and RHS of the equation are quantifying the likelihood of the $(H + t)^{th}$ timestep. This is because the state in RHS is from $t + 1^{th}$ timestep while the bin index is $H - 1$, effectively quantifying the likelihood of the $(H + t)^{th}$ timestep. This identity can be used

to derive a temporal consistency identity for distributional NCE, which is satisfied by the Bayes' optimal classifier (the solution to Eq. 4):

$$\frac{C^\pi(s_t, a_t, g)[H]}{1 - C^\pi(s_t, a_t, g)[H]} = \mathbb{E}_{(s_{t+1}, a_{t+1}) \sim (s_t, a_t)} \left[ \frac{C^\pi(s_{t+1}, a_{t+1}, g)[H-1]}{1 - C^\pi(s_{t+1}, a_{t+1}, g)[H-1]} \right]. \tag{13}$$

We now turn this identity into a penalty for the distributional NCE classifier as follows:

$$L_{TC} = \mathbb{E}_{(s_t, a_t, g, s_{t+1}, a_{t+1})} \Big[ \lfloor C(s_{t+1}, a_{t+1}, g)[H-1] \rfloor \log C(s_t, a_t, g)[H]$$

$$+ \lfloor (1 - C(s_{t+1}, a_{t+1}, g)[H-1]) \rfloor \log (1 - C(s_t, a_t, g)[H]) \Big], \tag{14}$$

where $\lfloor . \rfloor$ denotes the stop-gradient operator. Because the identity in Eq. 13 is satisfied by the Bayes' optimal classifier, adding the corresponding penalty (Eq. 14) to the distributional NCE loss (Eq. 3) does not change the solution.

### C.1 MULTI-STEP TEMPORAL CONSISTENCY

In Sec. 5.2, we detailed the temporal consistency identity in Eq. 13 and proposed a 1-step temporal consistency regularization objective in Eq. 14 to enforce it. We also briefly introduced a $k$-step extension of this objective in Eq. 7. In this section, we formally derive the $k$-step consistency objective.

**Deriving the multi-step consistency regularization.** Let $p^{\pi(\tau)}(s_t, a_t, g, H)$ be the distribution over $H$-length state-action trajectories generated by the goal-conditioned policy $\pi_g = \pi(.|., g)$ with $s_t$ as the start state and $a_t$ as the first action. Then, the future state probabilities under $\pi_g$ satisfy the following identity:

$$p^{\pi_g}(g \mid s_t, a_t, H) = \mathbb{E}_{(s_{t+1}, a_{t+1}, s_{t+2}, a_{t+2}, \ldots, s_{t+H-1}, a_{t+H-1}) \sim p^{\pi(\tau)}(s_t, a_t, g, H-1)} \left[ p(g \mid s_{t+H-1}, a_{t+H-1}) \right]$$

$$= \mathbb{E}_{s_{t+1} \sim p(.|s_t, a_t), a_{t+1} \sim \pi(.|s_{t+1}, g)} \Big[ \mathbb{E}_{(s_{t+2}, a_{t+2}, \ldots, s_{t+H-1}, a_{t+H-1}) \sim p^{\pi(\tau)}(s_{t+1}, a_{t+1}, g, H-2)}$$

$$\left[ p(g \mid s_{t+H-1}, a_{t+H-1}) \right] \Big]$$

$$= \mathbb{E}_{s_{t+1} \sim p(.|s_t, a_t), a_{t+1} \sim \pi_g(.|s_{t+1})} \left[ p^{\pi_g}(g \mid s_{t+1}, a_{t+1}, H-1) \right]. \tag{15}$$

This identity can be enforced over the distributional classifier using the 1-step temporal consistency regularization objective in Eq. 14. However, this property also holds for $k > 1$ steps:

$$p^{\pi_g}(g \mid s_t, a_t, H) = \mathbb{E}_{(\ldots, s_{t+k}, a_{t+k}) \sim p^{\pi(\tau)}(s_t, a_t, g, k)} \left[ p^{\pi_g}(g \mid s_{t+k}, a_{t+k}, H-k) \right]. \tag{16}$$

We use the identity in Eq. 16 to derive a temporal consistency identity for distributional NCE. This identity is satisfied by the Bayes' optimal classifier (the solution to Eq. 4)[2]:

$$\frac{C^\pi(s_t, a_t, g)[H]}{1 - C^\pi(s_t, a_t, g)[H]} = \mathbb{E}_{(\ldots, s_{t+k}, a_{t+k}) \sim p^{\pi(\tau)}(s_t, a_t, g, k)} \left[ \frac{C^\pi(s_{t+k}, a_{t+k}, g)[H-k]}{1 - C^\pi(s_{t+k}, a_{t+k}, g)[H-k]} \right], \tag{17}$$

and then turn this identity into an auxiliary, consistency objective:

$$L_{TC}^k = \mathbb{E}_{(s_t, a_t, g, s_{t+k}, a_{t+k})} \Big[ \lfloor C(s_{t+k}, a_{t+k}, g)[H-k] \rfloor \log C(s_t, a_t, g)[H]$$

$$+ \lfloor (1 - C(s_{t+k}, a_{t+k}, g)[H-k]) \rfloor \log (1 - C(s_t, a_t, g)[H]) \Big]. \tag{18}$$

The consistency objective above is valid for any goal-conditioned policy $\pi(.|., g)$, as long as $(s_{t+k}, a_{t+k})$ is the $k^{th}$ intermediate step on the Markov chain generated by the policy that connects $s_t$ and $g$. In our practical implementation, we sample $(s_t, a_t, s_{t+k}, a_{t+k}, s_{t+H})$, $k < H$, from the replay buffer, and relabel the goal $g = s_{t+H}$ to train the critic via direct contrastive loss (Eq. 3) and $k$-step temporal consistency regularization (Eq. 18). As a result, the critic estimates the future state density of the average hindsight-relabeled policy over the replay buffer rather than the current policy, just like prior MC contrastive RL algorithms (Eysenbach et al., 2022). In our implementation, the

---

[2]We use $C^\pi(s, a, g)$ to denote the Bayes' optimal classifier for a policy $\pi(a|s, g)$.

future goal distance $H$ is a random variable sampled from a Geometric distribution $H \sim \text{GEOM}(\gamma)$, and the intermediate state distance $k$ is sampled from a truncated distribution to enforce that $k < H$. We call this method "Distributional NCE with Multi-step temporal consistency regularization." Like temporal difference methods (Eysenbach et al., 2020), the temporal consistency regularization enables information and uncertainty over future states to flow back in time, thereby accelerating the Monte-Carlo classifier training.

**Handling the edge-case: Consistency update for the catch-all bin.** When applying the $k$-step temporal consistency loss, the catch-all bin gets mapped to $k + 1$ bins from the future state, unlike regular classifier bins that have a $1 : 1$ mapping with a corresponding future classifier bin. This is an artifact of using a finite number of bins to represent the infinite-horizon discounted probabilities. More precisely, the equivalent temporal consistency identity (Eq. 17) for the catch-all bin, which is satisfied by the Bayes' optimal classifier (the solution to Eq. 4):

$$\frac{C^\pi(s_t, a_t, g)[h]}{1 - C^\pi(s_t, a_t, g)[h]} =$$

$$\mathbb{E}_{(s_{t+k}, a_{t+k})} \left[ \sum_{i=1}^{h-1} \left( (1-\gamma)\gamma^{i-1} \frac{C^\pi(s_{t+k}, a_{t+k}, g)[i-k]}{1 - C^\pi(s_{t+k}, a_{t+k}, g)[i-k]} \right) + \gamma^{h-1} \frac{C^\pi(s_{t+k}, a_{t+k}, g)[h-k]}{1 - C^\pi(s_{t+k}, a_{t+k}, g)[h-k]} \right],$$

where $h$ is the total number of classifier bins and also the index of the catch-all bin. This identity can then be turned into a penalty as follows:

$$L_{TC}^k = \mathbb{E}_{(s_t, a_t, g, s_{t+k}, a_{t+k})} \Big[ \lfloor w(s_{t+k}, a_{t+k}, g) \rfloor \log C(s_t, a_t, g)[H]$$

$$+ \lfloor (1 - w(s_{t+k}, a_{t+k}, g)) \rfloor \log (1 - C(s_t, a_t, g)[H]) \Big],$$

where $w(s, a, g) = \sum_{i=1}^{h-1} \left( (1-\gamma)\gamma^{i-1} \frac{C(s,a,g)[i-k]}{1-C(s,a,g)[i-k]} \right) + \gamma^{h-1} \frac{C(s,a,g)[h-k]}{1-C(s,a,g)[h-k]}$.

# D ADDITIONAL EXPERIMENTS

## D.1 COMPARISON WITH PRIOR (NON-CONTRASTIVE) GCRL BASELINES

In Fig. 6, we report the performance of the Distributional NCE algorithm with prior (non-contrastive) GCRL baselines. Our baselines consist of (i) TD3+HER (Andrychowicz et al., 2017): a performant off-policy actor-critic algorithm combined with hindsight relabeling, (ii) goal-conditioned supervised learning (GCSL) (Ghosh et al., 2019): iteratively behavior clones the hindsight relabeled policy, and (iii) a model-based baseline: a goal-conditioned implementation of MBPO (Janner et al., 2019). We do not include an reward shaping experiments since prior work (Andrychowicz et al., 2017) has already shown has shown that they perform quite poorly (<10% success rate in most tasks) on robotic manipulation tasks, even when combined with hindsight relabeling.

## D.2 EXPLORING THE LOSS LANDSCAPE OF DISTRIBUTIONAL CLASSIFIERS

Prior works (Bellemare et al., 2017; Sun et al., 2022) have identified that distributional RL methods enjoy stable optimization and better learning signal compared to their counterpart RL methods. In particular, Sun et al. (2022) demonstrates that distributional value function approximations have a desirable smoothness property during optimization, which is characterized by small gradient norms. In this section, we try to examine if using the proposed distributional NCE algorithm enjoys some of these benefits. Note that prior works use the distributional critic to estimate the continuous return distribution with discretized bins (Bellemare et al., 2017; Dabney et al., 2018), which is different from our work that estimates the distributional probabilities of reaching the goal at discrete future timesteps.

In Fig. 8, we visualize the training loss and the gradient norm for the actor and critic networks over the course of training when optimized with the contrastive NCE and Distributional NCE algorithms. We note that the training loss for the critic network remains nearly unchanged, and the gradient norm is slightly smaller when switching from contrastive NCE to the Distributional NCE objective. On the other hand, we observe that actors trained with distributional critics receive gradients with smaller

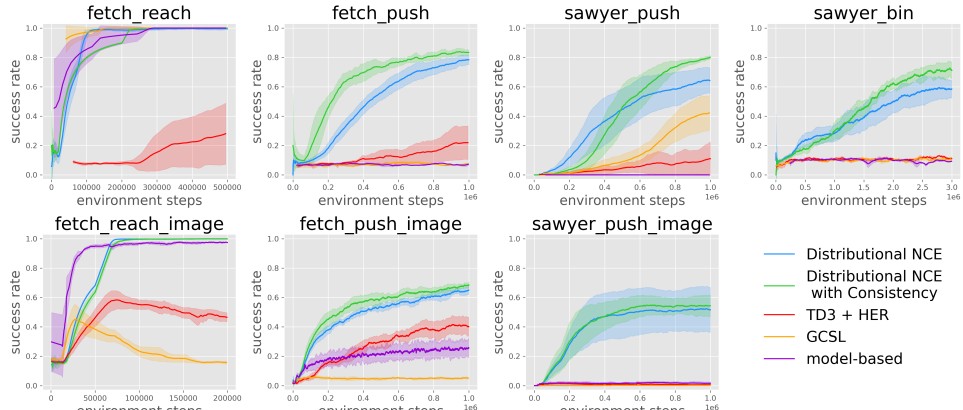

Figure 6: **Comparison with (non-contrastive) GCRL baselines.** Distributional NCE outperforms these baselines in all but the easiest tasks (fetch_reach, fetch_reach_image).

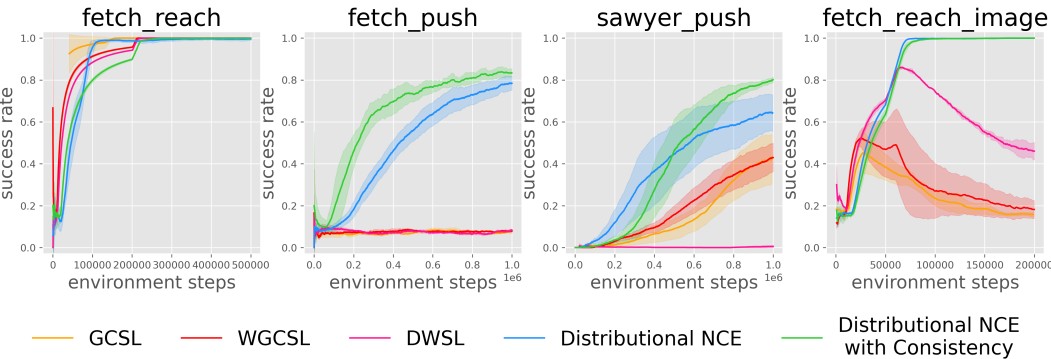

Figure 7: **Comparison with prior goal-conditioned supervised learning methods.** Distributional NCE outperforms GCSL (Ghosh et al., 2019), weighted GCSL (WGCSL) (Yang et al., 2022), and distance-weighted supervised learning (DWSL) (Hejna et al., 2023) in a majority of the tasks.

norms and achieve an overall lower loss. Note that plots in Fig. 8 are not a fair comparison since the Distributional NCE and Contrastive NCE agents were trained on different data, one that was collected by their respective actors interacting with the environment. However, we find the consistently low actor loss and smaller actor gradient norms with Distributional critics as compelling evidence to inspire future research works to study these optimization advantages more rigorously.

### D.3 COMPARISON WITH THE LAST-LAYER ENSEMBLE BASELINE

The Distributional NCE algorithm uses a distributional critic with $h$ classifier bins, while the Contrastive NCE (Eysenbach et al., 2022) uses a regular critic with 1 output bin to directly denote the probability of reaching the goal in the future (up to proportionality). For the distributional critic, we simply modified the last linear layer in the regular critic network to have $h$ outputs in all our experiments. In this section, we examine the importance of the distributional critic architecture by training a distributional critic with the Contrastive NCE algorithm. We do this by treating the distributional critic as an ensemble of critic networks, with all but the last-layer parameters shared.

We report the performance of the last-layer ensemble baseline in comparison to Contrastive NCE and Distributional NCE algorithms on the fetch_push task in Fig. 9. We observe that the last-layer ensemble baseline outperforms the Contrastive NCE algorithm by $+10\%$ higher success. It can also be seen that the Distributional NCE algorithm outperforms this ensemble baseline by $+7\%$. Further, Distributional NCE with consistency loss offers a $+15\%$ improvement over the ensemble baseline. This experiment confirms that the algorithm used to train the distributional critic has a huge impact on

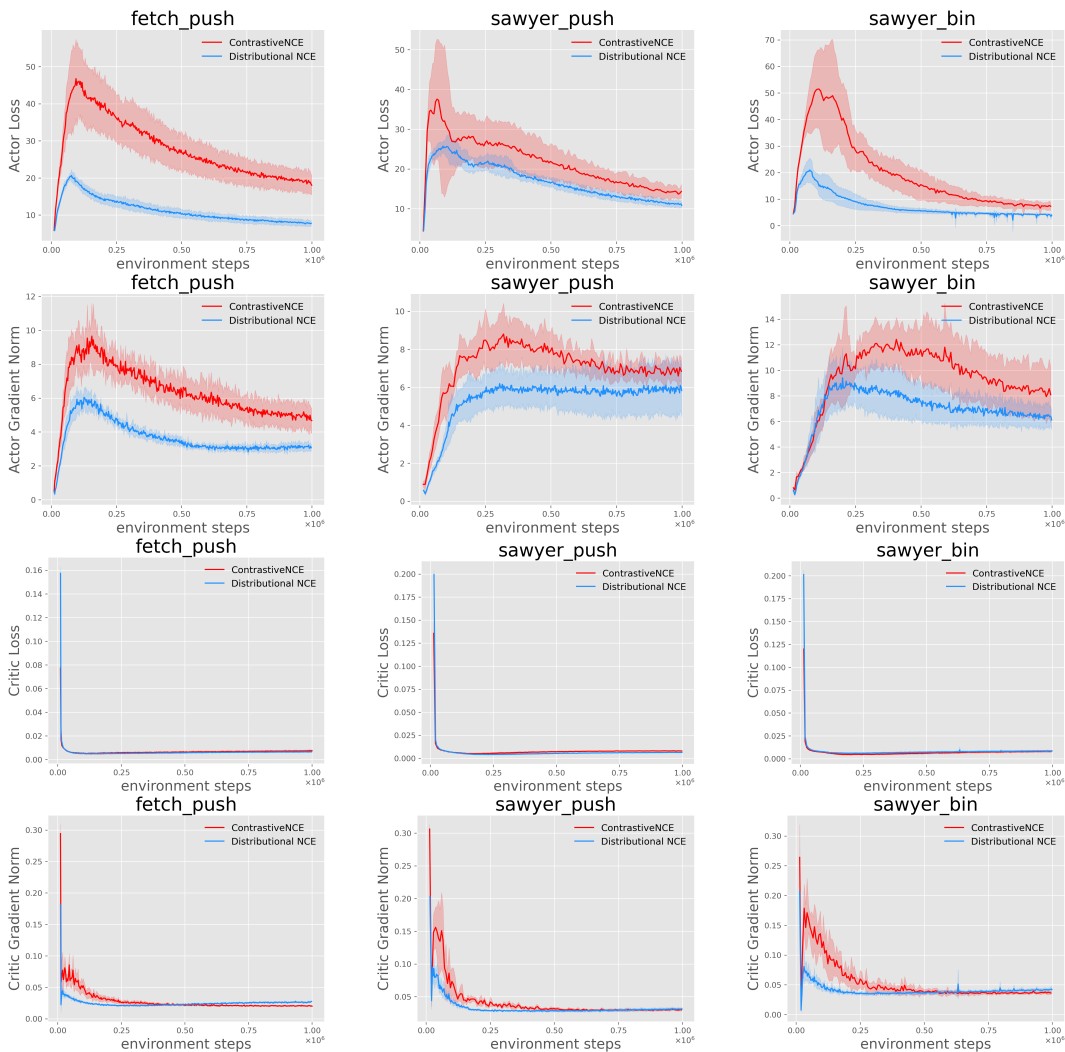

Figure 8: Visualization of the training losses and gradient norms for the actor and critic networks over the course of training. We do not see a huge difference in critic loss or gradients but observe that the actor loss is consistently lower and has a smaller gradient norm for Distributional NCE relative to Contrastive NCE.

the overall performance: Distributional NCE with Consistency > Distributional NCE > Contrastive NCE.

## D.4 PERFORMANCE IN STOCHASTIC ENVIRONMENTS

In Fig. 10, we report the performance of the Distributional NCE algorithm and MC distance regression in the Stochastic 2D maze environment. We observe that istributional NCE outperforms MC distance regression by over 20% higher success rate.

## D.5 HOW WELL DOES CONSISTENCY REGULARIZATION WORK IN PRACTICE?

We empirically found that 1-step temporal consistency regularization does not improve the performance of Distributional NCE; occasionally it decreases performance. On the other hand, we found that the multi-step temporal consistency regularization significantly boosts the performance of Distributional NCE in Fig. 11.

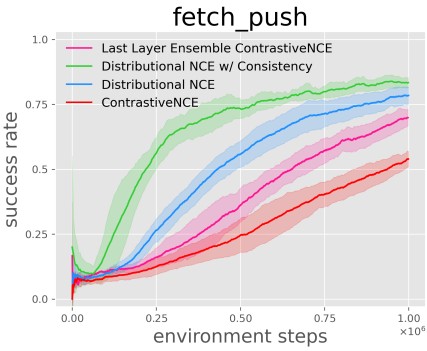

Figure 9: Distributional Critic trained with Contrastive NCE (last layer ensemble baseline) does not match Distributional NCE, highlighting the importance of the training algorithm over architectural choice.

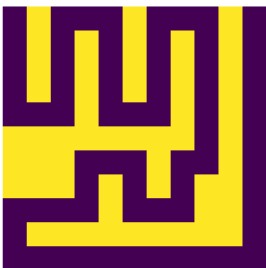
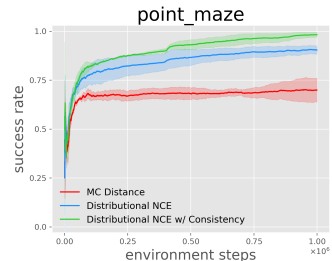

Figure 10: **Stochastic 2D maze environment**: (a) Maze layout (left) and (b) Distributional NCE outperforms MC distance regression by over 20% higher success rate (right). The agent's start and goal locations are chosen randomly in the valid regions (purple). The action space is $[-1, 1] \times [-1, 1]$, suggesting the displacement to the next state. The agent is not allowed to go through walls (yellow). To simulate stochasticity, we corrupt the action with a uniform random noise $\mathcal{N}(0, 0.5) \times \mathcal{N}(0, 0.5)$.

### D.6 PERFORMANCE OF DISTRIBUTIONAL NCE WITH DIFFERENT NUMBER OF BINS

In this section, we study if the choice of the number of classifier bins has an impact on the performance of the Distributional NCE algorithm. In theory, this hyperparameter should have no effect on convergence to the optimal policy since any Bayes-optimal distributional classifier can be mapped to the Bayes optimal contrastive NCE classifier (num_bins=1) as shown in Eq. 12. However, in practice, we find setting $H$ to be sufficiently large helps with improving the performance. For instance, when $H = 1$, Distributional NCE reduces to Contrastive NCE (Eysenbach et al., 2022), which does not perform as well (see Fig. 4). On the other hand, we also observed very little difference in performance for larger value of $H$. More precisely, on the sawyer_push task in Fig. 12, wherein we see very little difference in performance for four distinct choices for the number of classifier bins: 11, 21, 51, and 101. For all the rest of the experiments in the paper, we fix the number of classifier bins to 21.

### D.7 CONNECTION BETWEEN DISTRIBUTIONAL NCE AND THE "DISTANCE CLASSIFIER"

Our final visualization draws a connection between distributional NCE and the "Distance Classifier" baseline from Fig. 3. We can recover the Bayes' optimal distance classifier from the distributional NCE predictions by normalizing the predicted probabilities (Eq. 6). We visualize a confusion matrix of these predictions in Fig. 13 by averaging over 1000 states sampled from a trained policy. We observe that there is a clear trend along the diagonal, indicating that the distributional NCE predictions (after normalization) can be used to estimate "distances." This visualization not only provides a further sanity check that distributional NCE makes reasonable predictions, but also highlights that (by normalization) distributional NCE retains all the capabilities of distance prediction.

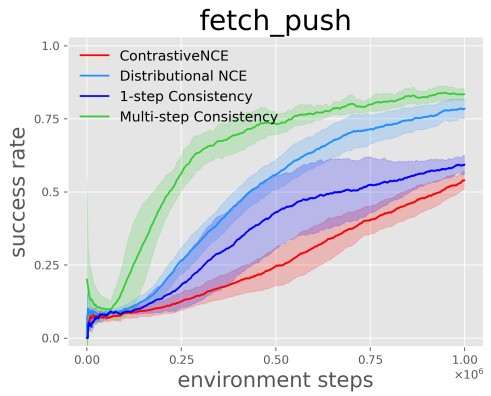 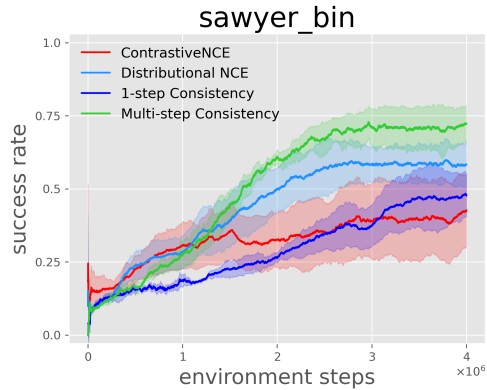

Figure 11: Multi-step temporal consistency regularization is significantly more effective than 1-step consistency regularization. In some cases, 1-step consistency regularization actually hurts the performance of the Distributional NCE algorithm, but Multi-step consistency almost always improves the performance.

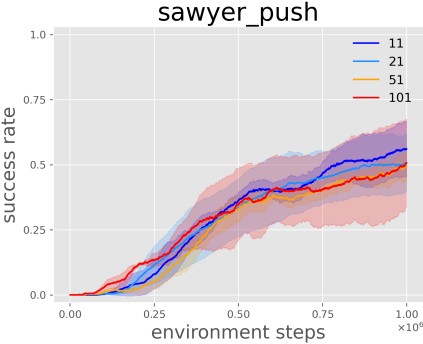

Figure 12: Varying the number of classifier bins has little effect on the performance of Distributional NCE for sawyer_push task.

## E  IMPLEMENTATION DETAILS

We used the official contrastive RL codebase[3] (Eysenbach et al., 2022) in the JAX framework (Bradbury et al., 2018) to run the contrastive RL baselines: Contrastive NCE (Eysenbach et al., 2022) and C-Learning (Eysenbach et al., 2020). Moreover, we implemented the distributional NCE algorithms by modifying this codebase as follows:

1. Change the last layer in the critic's architecture to output $h$ bins (Alg. 2).

2. Change the Contrastive NCE objective to the Distributional NCE objective (Alg. 1).

3. Add the consistency loss (Eq. 7,18) to the classifier training module.

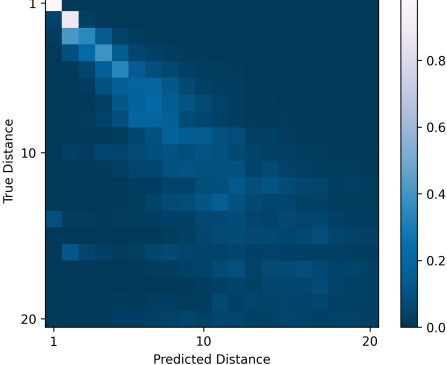

Figure 13: The predictions from distributional NCE can be converted into a distance classifier by normalization. See text for details.

The actor is trained using the actor loss from soft actor-critic (Haarnoja et al., 2018), while the critic is optimized for the contrastive classification objective in Eq. 3. In all our experiments, we report the mean performance and the 95% confidence interval computed across 5 random seeds. We ran all our experiments on a single RTX 2080 Ti GPU with 11GB memory.

[3]https://github.com/google-research/google-research/tree/master/contrastive_rl

| Hyperparameter | Value |
|---|---|
| number of classifier bins ($h$) | 21 |
| batch_size | 256 |
| EMA target network ($\tau$) | 0.005 |
| discount ($\gamma$) | 0.99 |
| hidden dims (policy and critic representations) | (256, 256) |
| critic representation dimension | 64 |
| learning rate | 3e-4 |
| Optimizer | Adam (Kingma & Ba, 2014) ($\beta_1 = 0.9, \beta_2 = 0.999$) |
| goal relabelling ratio for actor loss | 0.5 |
| maximum replay buffer size | 1,000,000 |
| minimum replay buffer size | 10,000 |

Table 1: A list of important hyperparameters used in our method and the baselines.

**Network Architecture**: We use the same architecture as the Contrastive NCE baseline while modifying the last layer in the critic. The policy is a standard 2-layer MLP with ReLU activations and 256 hidden units. The critic network comprises of a state-action encoder and a goal encoder (Alg. 2), which are each 2-layer MLP with ReLU activations and 256 hidden units, and a final dimension of $repr\_dim \times h$ ($repr\_dim = 64$ and $h = 21$ in all our experiments). For image-based tasks, we use the standard Atari CNN encoder (Mnih et al., 2013; Eysenbach et al., 2022) to project the state and goal image observations into the latent space before passing them into the policy and critic networks.

**Hyperparameters**: We keep the default hyperparameters of Contrastive NCE (Eysenbach et al., 2022) for all our experiments (Table 1). The proposed Distributional NCE algorithm only introduces one extra hyperparameter - the number of classifier bins, which is set to 21 in all the experiments.

### E.1 DISTRIBUTIONAL CRITIC IMPLEMENTATION

In this section, we go over the pseudo-code to implement a distributional critic network with $h$ classifier bins in Alg. 2. The output of the distributional critic is a ternary tensor with the first two axes corresponding to the state-action and goal indices, and the last axis $h$ is the classifier bin index. The main diagonal along the first two axes corresponds to positive examples, i.e., state-action representations paired with their corresponding future states (reachable goals). Every other off-diagonal term corresponds to a negative example, i.e., a state-action representation paired with a randomly sampled goal.

**Algorithm 2** DISTRIBUTIONAL CLASSIFIER: The contrastive classifier block, where the main diagonal corresponds to positive examples and off-diagonal entries correspond to negative examples. h is the number of bins in the classifier output, which may be less than the task horizon. Comments denote the shapes of tensors.

```python
def classifier(states, actions, goals):
  sa_repr = sa_encoder(states, actions)  # (batch_size, h, repr_dim)
  g_repr = g_encoder(goals)  # (batch_size, h, repr_dim)
  logits = einsum('ikl, jkl->ijk')  # (batch_size, batch_size, h)
  # logits[i, j, k] is the probability of going from s[i] to s[j] in k steps.
  return logits
```

### E.2 TASK DESCRIPTIONS

We conduct our experiments on four standard simulated robot manipulation tasks (Plappert et al., 2018; Yu et al., 2020) with increasing complexity: fetch_reach, fetch_push, sawyer_push, and sawyer_bin. All our tasks are framed as reward-free goal-reaching problems where the performance of the agent is tracked by the fraction of times it successfully reaches the goal.

**fetch_reach**: This task involves controlling a simulated fetch robotic arm to move the gripper to a specified 3D goal position. This is the simplest of all four tasks, where greedily moving the gripper toward the target position solves the task.

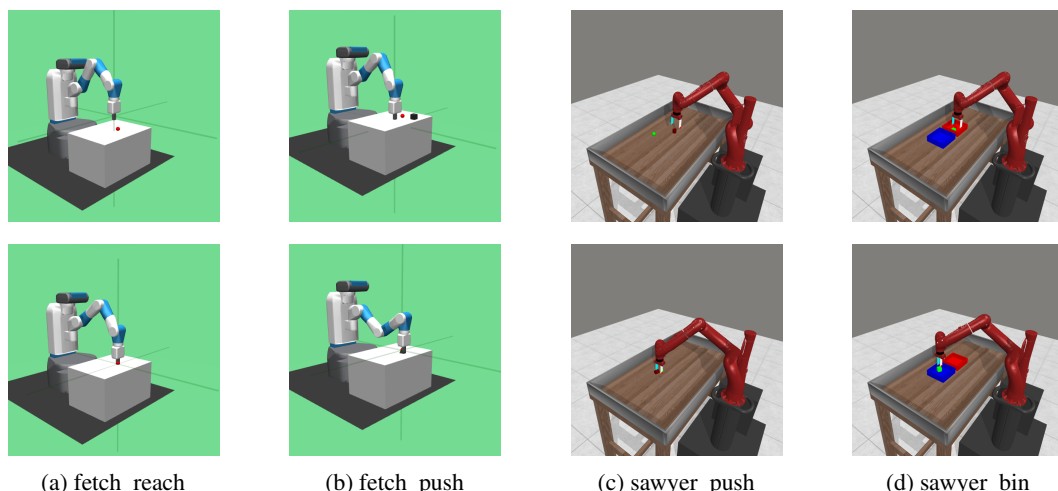

|  (a) fetch_reach  |  (b) fetch_push  |  (c) sawyer_push  |  (d) sawyer_bin  |

Figure 14: Illustration of the goal-reaching tasks used in this paper. The top row is a sample state at the time of initialization, and the bottom row is the corresponding goal state.

| Task | Observation Space (state and goal) | Action Space | Max Episode Length |
|---|---|---|---|
| fetch_reach | 20 | 4 | 50 |
| fetch_push | 50 | 4 | 50 |
| sawyer_push | 14 | 4 | 150 |
| sawyer_bin | 14 | 4 | 150 |
| fetch_reach_image | $64 \times 64 \times 6$ | 4 | 50 |
| fetch_push_image | $64 \times 64 \times 6$ | 4 | 50 |
| sawyer_push_image | $64 \times 64 \times 6$ | 4 | 150 |

Table 2: Environment details for the selected goal-reaching tasks.

**fetch_push**: In this task, the same simulated fetch robotic arm needs to push a block placed on the table to a specified position. This is a harder task since the agent needs to reason about the dynamics of precisely pushing a block to a specified location. The agent needs to be careful as it can enter unrecoverable states, such as the block falling off the table if pushed incorrectly. Note that the gripper fingers are disabled, in order to force the agent to push the block to the goal rather than pick-and-place it at the goal.

**sawyer_push**: This task is similar to fetch_push but involves controlling a simulated sawyer robotic arm. A key difference is that this is a longer horizon task with $3\times$ as many steps as fetch_push in each episode before termination.

**sawyer_bin**: In this task, the same simulated sawyer robotic arm needs to pick a block from a randomized position in one bin and put it in a goal location in another bin. This is a hard exploration problem since the agent must learn the skills associated with (i) picking and dropping an object and (ii) moving the gripper to a desired location, and learn to coordinate these skills in the pick-move-drop sequence to solve the task. Failing to do even one of these skills/sub-tasks correctly will result in an unsuccessful outcome.

We also conduct our experiments on the following image-based variants of the above-mentioned tasks: fetch_reach_image, fetch_push_image, and sawyer_push_image. In these tasks, the low-dimensional observation space is replaced with a $64 \times 64$ image. We chose these tasks to demonstrate that the Distributional NCE algorithm is able to estimate the probability of reaching the goal over future timesteps directly from image observations. To get a better idea of the tasks, we visualized a random start state and the corresponding goal state for each of these tasks in Fig. 14. Moreover, the dimensionality of the observation and action space is described in Table 2.

