# OpenReview forum: "Distributional Distance Classifiers for Goal-Conditioned Reinforcement Learning"
_ICLR.cc/2024/Conference — Submitted to ICLR 2024_

### Official Review · Reviewer_iQmu · 2023-10-25

**Soundness:** 3 good
**Presentation:** 3 good
**Contribution:** 3 good
**Rating:** 8
**Confidence:** 4

**Summary:**

This paper studies goal-conditioned reinforcement learning, and the authors propose a new method where they estimate the probability of reaching the goal at different time steps instead of just estimating the distance to the goal. Their final algorithm, distributional NCE, achieves promising empirical performance in several standard goal-conditioned environments.

**Strengths:**

Goal-conditioned RL is an important topic as it has many application scenarios. The idea proposed in this paper is quite natural and the author provide a good information theory argument on why their algorithm should work better (the learner receives logH bits information instead of 1 bits for each positive example). The resulting algorithm is simple, intuitive, and has promising empirical performance.

**Weaknesses:**

1. The writing is not very clear and require the reader to be familiar with previous work on goal-conditioned RL. For example, when introducing the objective of contrastive RL on page three, it is unclear what C(s, a, g) stands for.
2. I am not very convinced about the claim that "the number of bins does not affect the final performance".  In the extreme case, where H=1, it becomes a quantity similar to the expected distance measure and it should perform similar to estimating expected distance measure in my opinion. I would like to see how the algorithm behaves when H is very small, such as H=1 or 2.

**Questions:**

1. The problem formulation in this work is not exactly the stochastic shortest path problem as the discounting factor is < 1. Will the analysis still go through if the discounting factor is 1?
2. In related work, please also discuss the relationship between this work and the recent theoretical advancement in goal-conditioned RL, such as the followings:
(1) Jean Tarbouriech, Evrard Garcelon, Michal Valko, Matteo Pirotta, and Alessandro Lazaric. Noregret exploration in goal-oriented reinforcement learning. In International Conference on Machine Learning, pages 9428–9437. PMLR, 2020
(2) Alon Cohen, Haim Kaplan, Yishay Mansour, and Aviv Rosenberg. Near-optimal regret bounds for stochastic shortest path. In Proceedings of the 37th International Conference on Machine Learning, volume 119, pages 8210–8219. PMLR, 2020.
(3) Liyu Chen, Mehdi Jafarnia-Jahromi, Rahul Jain, and Haipeng Luo. Implicit finite-horizon approximation and efficient optimal algorithms for stochastic shortest path. Advances in Neural Information Processing Systems, 2021a.
3. The proposed method has the same spirit as distributional RL. I am wondering whether quantile regression can also be extended to solving goal-oriented RL?

---

> ### Author Response · Authors · 2023-11-17
>
> Thanks for the valuable feedback. We are pleased to hear positive remarks about the significance of our contributions. In what follows, we will try to answer the questions as best we can:
>
> > The problem formulation in this work is not exactly the stochastic shortest path problem as the discounting factor is < 1. Will the analysis still go through if the discounting factor is 1?
>
> The analysis in Sec. 4.2 and Appendix A shows that the (discounted) probability objective used in our work is closely related to the (undiscounted) stochastic shortest path problem. This analysis requires that (discounted) probability objective is a discount factor strictly less than one; we have revised Appendix A to note this. Please let us know if this doesn't make sense, or if we have misunderstood the question.
>
>
> > In related work, please also discuss the relationship between this work and the recent theoretical advancement in goal-conditioned RL
>
> We have added a paragraph to the related work section noting the similarities and differences with our work. We welcome additional suggestions for related work to discuss.
>
> > The proposed method has the same spirit as distributional RL. I am wondering whether quantile regression can also be extended to solving goal-oriented RL?
>
> This is a very exciting future direction that is beyond the scope of this paper. The contrastive RL framework offers flexibility to arbitrarily define the bins of the distributional NCE algorithm, as long as their union covers all the future timesteps [1, \infty). Instead of choosing bin boundaries as successive timesteps as we do in the paper (akin to C51), one can choose them to be the quantiles of a normalized distance classifier P(H | s, a, g) (akin to QR-DQN).
>
> > I would like to see how the algorithm behaves when H is very small, such as H=1 or 2
>
> In the case of H=1, Distributional NCE reduces to the Contrastive NCE baseline, which is worse than our method (see Fig. 4). However, we also observed that varying the number of bins between 11 and 101 did not significantly impact the performance in Appendix D.6 Fig. 12, which hints that increasing the number of bins beyond a point has diminishing improvements. We have revised the paper (Appendix D.6) to clarify that the performance our method does degrade for very small $H$.

---

> > ### Comment · Reviewer_iQmu · 2023-11-22
> > **Response to rebuttal**
> >
> > Thanks for the reply. I would like to keep my positive evaluation.

---

### Official Review · Reviewer_RXdj · 2023-10-31

**Soundness:** 2 fair
**Presentation:** 2 fair
**Contribution:** 3 good
**Rating:** 3
**Confidence:** 2

**Summary:**

The authors of this work propose an algorithm, Distribution NCE, for efficient learning in Goal-Conditioned RL. The authors notice that Monte Carlo distance functions are problematic (as they are do not obey triangle inequality) and propose a remedy that estimates the probabilities of reaching the goal state. The authors then demonstrate the numerical performance of their remedy on seven standard goal conditional RL environment.

**Strengths:**

The empirical results of the paper appear to be good. The authors are able to demonstrate good performance with their Distributional NCE method. However, I am not familiar with empirical works.

**Weaknesses:**

I am not sure if propositions 1-3 are correct, the statements themselves are quite vague. The proofs, or lack thereof, are not convincing. For instance what is the argument being made in the proof of proposition 2? That MC distance are not valid distances because they do not satisfy triangle inequality ? How is this being shown for any metric? What is the distance function $d(s_1,s_2,a)$. Is the proof for proposition 1 referring to the mdp in figure 1a) ?

The main weakness of the paper lies in the formalism of the mathematical writing. As pointed out, certain mathematical expressions such as the Monte Carlo Distance Function are defined in words but not formally stated. This becomes an issue when writing proofs about these ill-defined terms. It might be better to just have a section illustrating the failure case of Monte Carlo Distance Functions as opposed to writing "propositions". If the authors want to write propositions then please spend some effort defining all the quantities to be used in the propositions.

**Questions:**

In the second equation on page 3, why is the expectation on the left of the sum different than the one on the right ? Also for equation (1) it seems the expectations are taken wrt to difference distributions?

What is a Mount Carlo Distance Function? Does it depend on the policy being played? Is there any expectations or does it just take in raw observations?

Why is Monte-Carlo distance estimation equivalent in the limit to learning a normalized distance classifier ?

COMMENTS:

When writing out theorem blocks, such as propositions, please include a proof block as well. In section 4.1 the "propositions" (1-3) have no proofs? However, proposition 4 seems to have a proof.

Appendix D.4 is empty.

---

> ### Author Response · Authors · 2023-11-17
>
> We thank the reviewer for the detailed review. It seems like the reviewer's main concern is about mathematical correctness and precision. We believe that all the theoretical results are correct, and have revised the paper to make the mathematical notion more clear (blue text). We answer specific questions below. **Do these revisions and answers fully address the reviewer's concerns?** We look forward to continuing the discussion.
>
> > I am not sure if propositions 1-3 are correct, the statements themselves are quite vague
>
> We believe that these propositions are correct, and have revised the statements to make the notation and results more clear. Please let us know if there is any notation is ambiguous or confusing, and we would be happy to further revise this section.
>
> > what is the argument being made in the proof of proposition 2?
>
> *Proposition 2* argues that the MC distance is generally not a valid metric. The proof is a proof by contradiction, so we construct an example where the MC distance is not a valid metric (i.e., it violates the triangle inequality). We have added a note at the start of the proof to highlight that it is a proof by contradiction.
>
> > Is the proof for proposition 1 referring to the mdp in figure 1a?
>
> Yes, the proof for *Proposition 1* is referring to the mdp in Fig. 1(a). We have revised the proof to clarify this point.
>
> > What is a Mount Carlo Distance Function? Does it depend on the policy being played? Is there any expectations or does it just take in raw observations?
>
> We added a formal definition of the Monte Carlo distance function at the start of Sec. 4. Monte Carlo (MC) distance function **d(s,a,g)** takes as input state-action-goal tuple (s, a, g) and outputs the expected number of steps a policy takes to go from a start state **s** and action **a** to a goal state **g**. As the name suggests, it is computed from Monte-Carlo trajectory rollouts of the policy. MC distance thus depends on the policy and is an expectation.
>
> > In the second equation on page 3, why is the expectation on the left of the sum different than the one on the right ?
>
> In this equation ($\arg\min_C \cdots$), the expectations are different because contrastive learning entails sampling positive and negative examples from different distributions. The first expectation in the equation is under a positive distribution (goal states that occur in the future timesteps of (s,a)) with a training label=1 and the second expectation is under the negative distribution (random goal states that have nothing to do with (s,a)) with a training label=0.
>
> > Why is Monte-Carlo distance estimation equivalent in the limit to learning a normalized distance classifier?
>
> We have revised Sec. 4.3 to clarify that these are not exactly equivalent. MC distance function can be estimated using a normalized distance classifier as follows:
> - MC distance d(s,a,g) = \sum_{i=0}^\infty ( i * P(i | s, a, g) ),
>
> where P(H | s, a, g) is the normalized distance classifier.
>
> > Appendix D.4 is empty
>
> We have added a sentence at the start of Appendix D.4 to clarify that this subsection contains just Fig. 10 and the text in its caption.

---

> > ### Author Response · Authors · 2023-11-20
> >
> > Dear Reviewer,
> >
> > We hope that you've had a chance to read our responses and clarifications. **Do the revisions and clarifications fully address the concerns?** As the end of the discussion period is approaching, we would greatly appreciate it if you could confirm that our updates have addressed the concerns, or further clarify the concerns!
> >
> > Kind regards,
> > The Authors

---

### Official Review · Reviewer_A8Ng · 2023-10-31

**Soundness:** 3 good
**Presentation:** 3 good
**Contribution:** 2 fair
**Rating:** 5
**Confidence:** 5

**Summary:**

Goal Conditioned Reinforcement Learning requires the agent to reach the goal state a minimum number of timesteps. A number of methods estimate the distances between two states and use this estimate to select the optimal action. However, the authors show why such a measurement is incorrect. They build upon prior methods for estimating future state visitation densities and come up with an algorithm that estimates the probability of reaching the goal in future t steps. Using the probability of the goal being reachable in H steps, they update their policy.

**Strengths:**

(1) They correctly highlight the shortcomings of the MC distance regression methods.

(2) They draw interesting insight into the relationship between maximum likelihood and stochastic shortest path.

(3) Their algorithm improves over the previous baseline Contrastive NCE on competitive domains.

**Weaknesses:**

(1) A common belief in the paper is that the natural way of designing a goal-conditioned RL problem is by minimizing the hitting time i.e. having rewards = -1 for all states and 0 at the goal. I believe it is more common to consider reward = 0 everywhere else and 1 at the goal as defined by the original problem. Subtracting -1 from all rewards acts like shaping the reward function which might lead to suboptimality as already shown in several works.

(2) Another assumption is that the method of predicting the number of steps that elapse between one observation and another is common. A few works definitely explore this direction, but I think it is more common to simply use a shaping reward over the 0/1 reward function.

(3) In section 4.1, it is unclear in the toy examples when the episode ends. From the math it seems like the episodes end when the goal state is reached otherwise they continue indefinitely.

(4) It would be interesting to see how this method compares with shaping rewards.

**Questions:**

(1) Why do MC distance regression methods estimate the Q function? Why can't I use the estimated distance as a reward? Using them as rewards is still valid even when they are not quasimetric.

(2) In section 4.3, the estimated distance functions seem to be odd. Suppose, C(s, a, g)[H] = [0, 0, ...]. Will the estimated distance between s and g be 0? Also, if the classifer predicts normalized probabilities, then p(g|s, a2, H=1, 2, 3, ...) cannot be equal to [1, 1, 1, ...].

(3) The algorithm is not completely clear. What is dt in the algorithm box?

---

> ### Author Response · Authors · 2023-11-17
>
> We thank the reviewer for the detailed review. It seems like the reviewer's main concern is about (i) the discrepancy in reward functions used in MC distance and contrastive RL frameworks and (ii) including a comparison with prior GCRL methods that perform reward shaping. We have revised the paper to include the reviewer’s suggestions (blue text). We answer specific questions below. **Do these revisions and answers fully address the reviewer's concerns?** We look forward to continuing the discussion.
>
> > It would be interesting to see how this method compares with shaping rewards.
>
> Prior work has shown that reward shaping performs quite poorly on these Fetch environments, even when combined with HER (https://arxiv.org/pdf/1707.01495.pdf, Fig 5, success rates all <10%). We have revised the additional experiments section (Appendix D.1) to note this.
>
> > Subtracting -1 from all rewards acts like shaping the reward function which might lead to suboptimality as already shown in several works.
>
> We agree that, in practice, RL methods that maximize a 0/1 return can sometimes work better than those that maximize a -1/0 return (despite these two objectives being mathematically equivalent in the discounted, infinite-horizon setting). Note that none of our baselines explicitly use the -1/0 reward function; rather, they either (i) estimate some form of distance/distribution directly from reward-free data or (ii) use temporal difference learning with a 0/1 reward (e.g. TD3+HER, MBRL baseline in Fig. 6). Our analysis discusses the -1/0 reward because it allows us to draw a connection between the standard RL problem (reward maximization) and the standard stochastic shortest path problem (minimizing expected steps to goal).
>
> > It is more common to simply use a shaping reward over the 0/1 reward function.
>
> We have revised Sec. 2 to clarify that, while there are a number of methods that predict distance, it is also a common practice to use shaping reward. Nonetheless, we believe our analysis is useful because we highlight the intricacies and shortcomings of using MC distance functions, and how these shortcomings can be fixed by taking a probabilistic perspective of distances.
>
> > Why do MC distance regression methods estimate the Q function? Why can't I use the estimated distance as a reward?
>
> MC distance regression methods [2,3] often use the learned distance function to select actions, similar to how Q functions are used to select actions. However, our theoretical analysis (Proposition 2) show that learned distances are not valid Q functions – they don't correspond to the expected returns of a clear reward function, so selecting actions by minimizing these learned distances may not necessarily correspond to reward maximization.
>
> Prior work has used learned distances as reward [1]. However, from a theoretical perspective, this is a bit tricky, as it's unclear what _global_ objective these methods are optimizing. That is, what does the sum of these temporal distances mean? These prior methods can achieve excellent results. Our method builds on these prior works by offering them a stronger theoretical footing, which can improve results in certain stochastic settings (see Fig. 4 and Appendix Fig. 6 and 7).
>
> > In section 4.1, it is unclear in the toy examples when the episode ends.
>
> The episode ends when the agent reaches the goal state. Our revised paper on OpenReview includes this clarification.
>
> > Suppose, C(s, a, g)[H] = [0, 0, ...]. Will the estimated distance between s and g be 0?
>
> The MC distance is ill-defined in this case. In practice, this corresponds to the setting where we've never seen any trajectories between s and g, so we cannot estimate the number of steps between them. We've revised Sec. 4.3 in the paper to include example.
>
> > If the classifer predicts normalized probabilities, then p(g|s, a, H=1, 2, 3, ...) cannot be equal to [1, 1, 1, ...].
>
> Correct: a _normalized_classifier cannot equal [1, 1, 1, …]. p(g|s, a, H=i) refers to the probability of reaching the goal state g as opposed to some other state at i^th timesteps, and is thus a quantity in [0,1]. On the other hand, p(H=i |s, a, g) refers to the probability of reaching the goal g at i^th timestep, given that the policy reaches the desired goal at some future timestep. Thus,
> - \sum_i p(g | s, a, H=i) \neq 1
> - \sum_i p(H=i | s, a, g) = 1
>
> > The algorithm is not completely clear. What is dt in the algorithm box?
>
> `dt` refers to the relative time index of the future state which is used to index the corresponding bin of the Distributional NCE classifier. We have updated the algorithm block to include this information.
>
> [1] K. Hartikainen, et al. Dynamical distance learning for semi-supervised and unsupervised skill discovery. arXiv:1907.08225, 2019.
>
> [2] S. Tian, et al. Model-based visual planning with self-supervised functional distances. In ICLR, 2021.
>
> [3] D. Shah, et al. Ving: Learning open-world navigation with visual goals. In IEEE ICRA, 2021.

---

> > ### Author Response · Authors · 2023-11-20
> >
> > Dear Reviewer,
> >
> > We hope that you've had a chance to read our responses and clarifications. **Do the revisions and clarifications fully address the concerns?** As the end of the discussion period is approaching, we would greatly appreciate it if you could confirm that our updates have addressed the concerns, or further clarify the concerns!
> >
> > Kind regards,
> > The Authors

---

> ### Comment · Reviewer_A8Ng · 2023-11-20
>
> The responses do clarify some of the concerns. But some of the concerns still stand,
>
> (1) It is true that HER often does not work with shaping rewards but works like [1] have shown good results with shaping rewards and hindsight experience replay. The HER paper considers just a simple shaping reward of L2 distance, but there can be other forms of shaping rewards too.
>
> (2) Are you sure 0/1 and -1/0 are mathematically equivalent?
>
> (3) The distance term $d^\pi(s, a, g)$ can very well be used as rewards. If not as a reward, then definitely as a potential-based shaping reward.
>
> [1] Durugkar et. al., Adversarial Intrinsic Motivation for Reinforcement Learning
>
> I believe that the work does show an interesting perspective of using probabilistic distances rather than MC distances. But, some additional studies are required as there are other ways in which distances are incorporated into policy learning. So, I stand by my rating.

---

> > ### Author Response · Authors · 2023-11-21
> >
> > Dear Reviewers,
> >
> > Thanks for clarifying the concerns! In what follows, we will try to answer the reviewer’s concerns as best we can:
> >
> > > I believe that the work does show an interesting perspective of using probabilistic distances rather than MC distances. But, some additional studies are required as there are other ways in which distances are incorporated into policy learning.
> >
> > We agree with the reviewer that there are many ways “MC distances” can and have been used in RL. While some approaches use them directly as Q-functions [2,3,4,5], others have used them as the reward function [6] or for reward shaping [1]. However, we believe that the bar for acceptance is a novel method with a strong theoretical foundation, and not to explore every possible way that method could be used.
> >
> > In this paper, we ask an orthogonal question: Are these “MC distances” even valid distance functions? (refer Sec. 4 to see why they are not a valid distance). Next, we propose a probabilistic framework to reason about distances that overcome some difficulties of MC distance functions. Our proposed method not only has appealing theoretical properties (we prove convergence to the optimal policy in Sec. 5.1 and Appendix B.2) but also achieves strong empirical results (10% median improvement in success rate against the second-best baseline across 7 benchmark environments - results from Fig 3,4,6,7). In particular, we included comparisons with relevant GCRL baselines: probabilistic GCRL (C-learning, Contrastive NCE), distance-based (MC distance function, normalized distance classifier), temporal difference (TD3+HER, Model-based RL), and supervised learning baselines (GCSL, WGCSL, DWSL). Overall, we observed that the methods from the probabilistic GCRL framework outperform other “MC distance” and reward-based baselines in our robotic benchmark tasks. We agree that there are likely different and potentially better ways to use “MC distances,” but comparing with all of them is beyond the scope of this paper.
> >
> > > It is true that HER often does not work with shaping rewards but works like [1] have shown good results with shaping rewards and hindsight experience replay. The HER paper considers just a simple shaping reward of L2 distance, but there can be other forms of shaping rewards too.
> >
> > We agree with the reviewer that there may exist other reward-shaping methods that result in faster learning for specific RL problems. However, shaping the rewards can be challenging [7,8,9] as it often requires domain knowledge or expert demonstrations (inverse RL, RLHF, contrastive preference learning). Moreover, it is also important to develop RL algorithms that can learn from minimal supervision. Our proposed method falls in the latter category of GCRL methods that avoid explicit supervision from humans.
> >
> > > Are you sure 0/1 and -1/0 are mathematically equivalent?
> >
> > Both 0/1 and -1/0 reward functions converge to the same optimal policy in infinite-horizon discounted MDP ($\gamma < 0$).
> > This can be proved by showing that the sum of discounted rewards (aka value functions) of any infinite-length sequence only differ by a constant factor:
> > $\sum_t \gamma^t (r_t - 1) = \sum_t (\gamma^t * r_t) - \sum_t (\gamma^t) = \sum_t (\gamma^t * r_t) - \frac{1}{1-\gamma}$
> >
> > > The distance term $d^\pi(s,a,g)$ can very well be used as rewards. If not as a reward, then definitely as a potential-based shaping reward.
> >
> > We agree with the reviewer that $d^\pi(s,a,g)$ can be used as a reward [6] or a potential function for shaping the rewards. We have revised the paper to note this.
> >
> > [1] Durugkar et. al., Adversarial Intrinsic Motivation for Reinforcement Learning
> >
> > [2] Tian et al., Model-based visual planning with self-supervised functional distances. In
> > ICLR, 2021.
> >
> > [3] Shah et al., Ving: Learning open-world navigation with visual goals. In IEEE ICRA, 2021.
> >
> > [4] Hejna et al., Distance weighted supervised learning for offline interaction data. In ICML, 2023.
> >
> > [5] Chen et al., Implicit finite-horizon approximation and efficient optimal algorithms for stochastic shortest path. In NeurIPS, 2021.
> >
> > [6] Hartikainen et al., Dynamical distance learning for semi-supervised and unsupervised skill discovery. arXiv preprint arXiv:1907.08225, 2019.
> >
> > [7] Skalse, et al., Defining and characterizing reward gaming. In NeurIPS, 2022.
> >
> > [8] Yuan, et al., A novel multi-step reinforcement learning method for solving reward hacking. Applied Intelligence 49 (2019): 2874-2888.
> >
> > [9] Everitt, et al., Reward tampering problems and solutions in reinforcement learning: A causal influence diagram perspective. Synthese 198.Suppl 27 (2021): 6435-6467.

---

### Official Review · Reviewer_CNkF · 2023-11-02

**Soundness:** 3 good
**Presentation:** 4 excellent
**Contribution:** 3 good
**Rating:** 8
**Confidence:** 2

**Summary:**

This paper constructs an example to illustrate that two criteria in goal-conditioned RL are not equivalent. It also uses Proposition 2 and 3 to show that Monte-Carlo distance is not a good criterion. It also designs an algorithm to train the agent w.r.t to latter criteria.

**Strengths:**

1. The observation that two criteria are not equivalent is interesting.

2. Section 4.1 justifies the use of probability instead of distance.

**Weaknesses:**

A lot of research in goal-conditioned RL uses probability as criterion. The paper should include a comparison to compare the algorithm ion the paper and previous algorithm.

**Questions:**

See 'Weakness' section.

---

> ### Author Response · Authors · 2023-11-17
>
> We thank the reviewer for the valuable feedback.
>
> > A lot of research in goal-conditioned RL uses probability as criterion. The paper should include a comparison to compare the algorithm in the paper and previous algorithm.
>
> Our experiments in Fig. 4 do compare with prior methods that use probability as a criterion (Contrastive NCE (red), C-Learning (yellow)). We'd welcome suggestions for additional baselines to compare against.

---

### Official Review · Reviewer_pEXE · 2023-11-08

**Soundness:** 2 fair
**Presentation:** 3 good
**Contribution:** 2 fair
**Rating:** 6
**Confidence:** 2

**Summary:**

This paper argues that optimizing for steps to a particular goal is not a feasible objective for most reinforcement learning problems. This work demonstrates the shortcomings of distance to goal metrics in toy Markov processes as well as experiments in larger state spaces. They then present an algorithm (Distributional NCE) that uses a probabilistic model of arriving at a goal in a given number of timesteps.

**Strengths:**

* Experimental questions and hypotheses were succinctly laid out and all appropriately addressed. The results adequately evidence the claims of 1) delivering a practical RL algorithm, 2) giving a probabilistic way of thinking about the probabilities and distances in a stochastic setting. This is achieved with a good mix of small and large problems.
* The work builds on relevant previous works while also explicitly stating its limitations.

**Weaknesses:**

**Technical Comments/Issues**
* The definition of Monte Carlo “distance” should be made clearer. The intuition is conveyed well, but since this concept is used for several examples in section 4.1 as well as in the experimental results, it would be useful to have some more mathematical grounding. Formally defining the state and state-action variant would be helpful to the reader. Something along the lines of:

i) d(s,g) = \mathbb{E}[\Delta | s_0=s, s_\Delta=g], where \Delta indicates the length of the trajectory and,

ii) d(s,g,a) = \mathbb{E}[\Delta | s_0=s, a_0 = a, s_\Delta=g]. This would also make the counter-example showing how this “distance” violates the triangle inequality more apparent.
* The temporal consistency objective does not make sense in certain reinforcement learning problems. Some goals can have a minimum number of steps needed before they can be reached. For example, in a gridworld where the shortest path to the goal is 10 steps away, P(reaching the goal in 9 steps) = 0, whereas P(reaching in 10) can be high for a good policy. It would be useful to address this aspect of temporal consistency or even highlight it in an experiment with such temporal constraints, or remove it as it does not naturally stem from the main hypothesis of the paper.
* Proposition 2 and 3 can be merged together. Proposition 2 is repeated in proposition 3, and the example showing how d(s,g) is not a quasi-metric can be explained alongside the example from V*
* While this work does acknowledge Contrastive NCE from the literature, the acronym is not explicitly spelled out anywhere. This would be useful to specify.

**Minor Comments/Issues**
* It would be useful to specify the number of runs that were averaged over in the learning curves of Figures 3, 4, and those in the appendix. The work should also mention what the shaded area around each curve represents and how they curves were smoothed, if any such technique was applied.
* In section 4.3, P^{\pi_g} was not defined. It would be useful to clarify it or remove it from the equation (as it only appears once in the main body).
* On page 11, Yecheng et al. 2022 was included twice in the references.

**Questions:**

* On Page 7, in the paragraph beginning “Comparison with distance regression.” it is claimed that “ We hypothesize that the stochasticity in action sampling from the policy, along with the associated risk of choosing the shortest path are ignored by MC distance functions [...]”. It is apparent that since the distance conditions itself on reaching the goal the risk of the shortest path is ignored, but the MC distance function is an average number of timesteps elapsed between a state and the goal state. Doesn’t it already account for the stochasticity in action selection?

---

> ### Author Response · Authors · 2023-11-17
>
> We thank the reviewer for the detailed review. It seems like the reviewer's main concern is about the temporal consistency objective and paper writing. We have revised the paper to include the reviewer’s suggestions (blue text). We answer specific questions below **Do these revisions and answers fully address the reviewer's concerns?** We look forward to continuing the discussion.
>
> > The temporal consistency objective does not make sense in certain reinforcement learning problems. Some goals can have a minimum number of steps needed before they can be reached. For example, in a gridworld where the shortest path to the goal is 10 steps away, P(reaching the goal in 9 steps) = 0, whereas P(reaching in 10) can be high for a good policy. It would be useful to address this aspect of temporal consistency or even highlight it in an experiment with such temporal constraints, or remove it as it does not naturally stem from the main hypothesis of the paper.
>
> Thanks for bringing up this great example; we believe the temporal consistency regularizer does make sense in this example. In this example, let’s imagine a start state s_0, a goal state g, and actions a_i ~ \pi(.|s_i) for all i \in [0, \infty). We further imagine that the goal state can only be reached after 10 steps:
> - P(s_0, s_{i}=g,a_0) = 0 for all i < 10 && P(s_0, s_{0 + “10”}=g,a_0) > 0.
> Nevertheless, the following still holds:
> - P(s_0, s_{0 + “10”}=g,a_0) = E_{s_1 ~ p(. | s_0, a_0), a_1 ~ \pi(.|s_1)} [ P(s_1, s_{1 + “9”}=g,a_1) ]
> Note that the term on the right-hand side refers to a future state that is 9 steps away relative to t=1, i.e. still the 10th timestep w.r.t the starting point t=0. We've revised the paper to include this discussion in Appendix C.
>
> > Does the MC distance function already account for the stochasticity in action selection?
>
> The MC distance function is conditioned on the current action, so it does not account  for the stochasticity in action selection at current timestep. However, it does account for the stochasticity in action selection at (finite) future timesteps since MC distance is a probabilistic average over the number of **finite** timesteps to the goal. However, MC distance discards the possibility of **never** reaching the commanded goal, in which case, the distance to the goal is infinite. In many real-world scenarios, there exists a non-zero probability of failing (a.k.a risk) to reach the goal, which is very important for correct decision-making. We show this in the toy MDP from Fig. 1(a) in the paper. Another example is an autonomous car that needs to cross the intersection but it is red light. MC distance suggests the vehicle to run the red light as it is the minimum-time path to the goal. Alternatively, our probabilistic framework accounts for the non-zero risk of collision with a passing vehicle upon running a red light, which would prevent the car from reaching its goal state, thereby making the right choice to wait until it’s green light.
>
> > specify the number of runs that were averaged over in the learning curves. The work should also mention what the shaded area around each curve represents and how they curves were smoothed, if any such technique was applied.
>
> The curves show the mean performance with the shaded region indicatingthe 95% confidence interval computed across 5 random seeds. We do not use any smoothing or filters to post-process the results. These details were in the original submission (Appendix E), but we have revised the main text to point readers to this appendix.
>
> > Minor comments
>
> Thanks for the suggestions to improve the paper writing! We have addressed these changes (in blue color) in the revised version of our paper on OpenReview:
> - Added a formal definition of the Monte Carlo distance function in Sec. 4
> - Revised Sec. 4.1 to better explain the pathological behaviors exhibited by MC distance functions.
> - Clarified that NCE stands for Noise Contrastive Estimation.
> - Removed duplicate reference to Yecheng et al. 2022 (page 11)

---

> > ### Author Response · Authors · 2023-11-20
> >
> > Dear Reviewer,
> >
> > We hope that you've had a chance to read our responses and clarifications. **Do the revisions and clarifications fully address the concerns?** As the end of the discussion period is approaching, we would greatly appreciate it if you could confirm that our updates have addressed the concerns, or further clarify the concerns!
> >
> > Kind regards,
> > The Authors

---

> > > ### Comment · Reviewer_pEXE · 2023-11-21
> > >
> > > Dear Authors,
> > >
> > > Many thanks for your revisions and clarifications, and I apologize for the time taken to read through the revised version.
> > >
> > > The author’s response clarifying temporal consistency aided my understanding. The addition to Appendix C on this matter is useful.
> > > Equation (1) is a useful addition.
> > > Corollary (1) is helpful in clarifying the nature of the MC “distance” used in this work.
> > > Spelling out NCE is appreciated!
> > >
> > > In light of this, I have updated the review to reflect these changes. Let me know if you ave any further concerns or questions regarding this assessment.
> > >
> > > Best wishes, Reviewer

---

### Meta-Review · Area_Chair_4X9A · 2023-12-11

**Metareview:**

This paper aims to present a valid distance metric for goal-based reinforcement learning to overcome issues with other metrics that do not satisfy the triangle inequality. While some reviews were positive, one reviewer pointed out flaws in the mathematical statements. After revision, at least one flaw remains that makes a proposition invalid. Specially, in Proposition 1, $d(1,a_1,4) = p + (1-p) \infty = \infty$ and not $1$ as stated in the proof. Due to this error and the possibility of others, the paper cannot be accepted.

**Justification For Why Not Higher Score:**

The mathematical errors need to be fixed, and based on the paper having prior errors and lack of unclear or missing definitions, it is possible that other errors may exist. As such, further revision and reviews are needed.

**Justification For Why Not Lower Score:**

N/A

---

### Decision · Program_Chairs · 2024-01-16

Reject